# Regional seismic risk assessment based on ground conditions in Uzbekistan

Vakhitkhan Alikhanovich Ismailov[1,2], Sharofiddin Ismatullayevich Yodgorov[1,3], Akhror Sabriddinovich Khusomiddinov [1,5], Eldor Maxmadiyorovich Yadigarov[1,3], Bekzod Uktamovich Aktamov[1,4], Shukhrat Bakhtiyorovich Avazov[1]

[1] Institute of Seismology of the Academy of Sciences Republic of Uzbekistan, Tashkent, 100128, Uzbekistan
[2] Tashkent State Technical University named after Islam Karimov, Tashkent, 100095, Uzbekistan
[3] National University of Uzbekistan, University Tashkent, University 4, 100174, Uzbekistan
[d] Tashkent University of Architecture and Construction, Tashkent, 100011, Uzbekistan
[5] Tashkent State Transport University, Tashkent, 100175, Uzbekistan

*Correspondence to*: Sh.Yodgorov (sh.i.yodgorov@gmail.com)

**Abstract.** The assessment of losses from strong earthquakes and the reduction of earthquake consequences are of great importance in maintaining the seismic safety. Special attention is given to evaluating the magnitude of economic losses caused by earthquakes, particularly the assessment of different levels of seismic risk, in order to protect the population and territories located in seismically active areas. To ensure sustainable development of countries, it is essential to estimate the economic losses that will occur in regions due to strong earthquakes and forecast them within the specified return periods at a given probabilities. Measures can then be implemented to mitigate the consequences of earthquakes.

For the basis of seismic risk assessment, maps of seismic intensity increment and an improved map of seismic hazard have been developed, taking into account the engineering-geological conditions of the territory of Uzbekistan and the seismic characteristics of soils.

For seismic risk map development, databases were created based on GIS platforms allowing us to systematize and evaluate the regional distribution of information on seismic hazards, number of buildings and construction types, coefficient of the seismic vulnerability of buildings, cadastral value of buildings, etc.

## 1. Introduction

As of January 1, 2022, the permanent population of Uzbekistan reached 35,271,276 people. Currently, approximately half of all Uzbekistan citizens (17.9 million people) live in urban areas and 17.4 million people live in rural areas[1]. At the territory of Uzbekistan and adjacent regions, both during the historical period and recent years, earthquakes with a local magnitude $M_L \geq$ 5 and an intensity at the epicenter $I_0$ reaching 6–10 according to the MSK-64 scale have been recorded (Table 1). In Table 1, earthquakes are listed whose epicenters are located near the specified city. It can be seen that many comparatively strong earthquakes have happened in Uzbekistan. Therefore, the problem of ensuring seismic safety at the territory of Uzbekistan is very relevant. The geological structure of Uzbekistan is very diverse, but the territory basically consists of two tectonic structures of the Tien Shan orogenic region and Turan plate. In the territory of Uzbekistan, tectonic movements are actively continuing nearly everywhere. In the geological history of Uzbekistan, throughout all stages of development, in particular, in the formation of the modern structural plan, faults, especially zones of deep faults, played an important role. These faults transect the entire Earth's crust, often penetrate into the mantle and are the natural boundaries of large structural elements. One of the challenges in assessing seismic risk involves considering the influence of soil conditions on the modification of seismic effects on the ground surface. Thus, a key objective of this study was to investigate the geological and lithological structure of the upper strata.

**Table 1**. Destructive earthquakes in the territory of Uzbekistan and adjacent territories. This data was retrieved from the database of the Institute of Seismology, Academy of Sciences of the Republic of Uzbekistan (2017).

| № | Date | | | Name of the nearest city/town | Latitude | Longitude | $M_L$ | Depth, km | Intensity, MSK-64 |
|---|------|-----|-------|-------------------------------|----------|-----------|-------|-----------|-------------------|
| | Year | Day | Month | | | | | | |
| 1 | 1868 | 3 | August | Tashkent* | 41,2 | 69,6 | 6,5 | 18 | VIII |
| 2 | 1883 | 14 | November | Osh* | 40,59 | 72,8 | 5,5 | 12 | VII |
| 3 | 1886 | 29 | November | Tashkent* | 41,4 | 69,5 | 6,0 | 14 | VIII |
| 4 | 1888 | 28 | November | Costakoz* | 40,2 | 69,3 | 5,6 | 10 | VIII |
| 5 | 1902 | 16 | December | Andijan* | 40,8 | 72,3 | 6,4 | 10 | IX |
| 6 | 1903 | 28 | March | Aimsk* | 40,8 | 72,69 | 6,1 | 14 | VIII |
| 7 | 1907 | 15 | September | Kyrkkol* | 40,3 | 72,5 | 5,8 | 10 | VIII |
| 8 | 1908 | 24 | March | Namangan* | 40,9 | 71,0 | 5,4 | 26 | VIII |

---

[1] *https://countrymeters.info/ru/Uzbekistan#population_densit*

| 9 | 1912 | 23 | January | Namangan* | 41,02 | 71,7 | 5,2 | 12 | VII-VIII |
|---|---|---|---|---|---|---|---|---|---|
| 10 | 1924 | 12 | July | Kurshabian* I | 40,5 | 73,1 | 6,4 | 25 | VIII |
| 11 | 1924 | 27 | July | Kurshab* II | 40,59 | 73,19 | 6,5 | 14 | IX |
| 12 | 1926 | 28 | May | Jalal-Abad* | 40,9 | 73,1 | 5,4 | 9 | VII-VIII |
| 13 | 1927 | 12 | August | Namangan* | 41,0 | 71,6 | 6,0 | 14 | VIII |
| 14 | 1929 | 18 | November | Chilean* | 41,5 | 63,5 | 5,2 | - | VIII |
| 15 | 1932 | 10 | February | Tamdybulak* | 41,3 | 65,2 | 6,1 | 25 | VII |
| 16 | 1935 | 5 | July | Boysun* | 38,3 | 67,4 | 6,2 | 16 | VIII |
| 17 | 1935 | 31 | May | Bulungur* | 39,6 | 67,1 | 5,4 | 20 | VII |
| 18 | 1937 | 18 | December | Pskem* | 42,1 | 70,9 | 6,4 | 17 | VIII |
| 19 | 1942 | 18 | January | Yartepa | 41,1 | 71,6 | 6,2 | 18 | VIII |
| 20 | 1946 | 3 | November | Chatkal | 41,9 | 72,0 | 7,5 | 25 | IX-X |
| 21 | 194 | 2 | June | Naiman | 40,9 | 72,3 | 5,9 | 9 | VII-VIII |
| 22 | 1955 | 19 | July | Bakhmal | 39,7 | 68,0 | 5,2 | 21 | VI-VII |
| 23 | 1959 | 25 | October | Burchmulla | 41,67 | 70,0 | 5,7 | 13 | VIII |
| 24 | 1965 | 17 | March | Koshtepa | 40,7 | 69,6 | 5,5 | 11 | VII |
| 25 | 1966 | 25 | April | Tashkent | 41,33 | 69,28 | 5,3 | 8 | VIII |
| 26 | 1968 | 13 | March | Kyzylkum I | 42,43 | 66,47 | 5,3 | 30 | VII |
| 27 | 1968 | 14 | March | Kyzylkum II | 42,59 | 66,45 | 5,0 | 30 | VII |
| 28 | 1968 | 8 | July | Baysun | 38,11 | 66,9 | 5,0 | 15 | VI-VII |
| 29 | 1970 | 19 | January | Pskent | 40,83 | 69,33 | 5,0 | 20 | VII |
| 30 | 1971 | 28 | October | Chatkal | 41,95 | 72,25 | 5,6 | 25 | VI-VII |
| 31 | 1976 | 8 | April | Ghazli I | 40,33 | 63,67 | 7,0 | 25 | IX |
| 32 | 1976 | 17 | May | Ghazli II | 40,28 | 63,38 | 7,3 | 20 | IX |
| 33 | 1977 | 19 | January | Isfara-Batken | 40,11 | 70,79 | 6,4 | 15 | VIII |
| 34 | 1977 | 21 | April | Khaidarkan | 40,11 | 70,95 | 5,7 | 14 | VII |
| 35 | 1977 | 6 | December | Tavaksai | 41,58 | 69,68 | 5,1 | 25 | VII |
| 36 | 1980 | 30 | December | Nazarbek | 41,33 | 69,05 | 5,5 | 12 | VIII |
| 37 | 1982 | 6 | May | Chimyon | 40,0 | 71,42 | 5,5 | 12 | VIII |
| 38 | 1984 | 17 | February | Papal | 40,22 | 71,5 | 5,6 | 14 | VIII |
| 39 | 1984 | 19 | March | Gazli | 40,38 | 63,36 | 7,2 | 15 | IX-X |
| 40 | 1985 | 28 | October | Kairakkum | 40,28 | 69,8 | 5,5 | 15 | VIII |
| 41 | 1987 | 26 | March | Altyntepa | 41,72 | 70,05 | 5,0 | 8 | VII |
| 42 | 1988 | 21 | December | Shamaldysai | 41,28 | 72,19 | 5,5 | 15 | VI-VII |
| 43 | 1992 | 15 | May | Izbazkent | 40,99 | 72,4 | 5,9 | 25 | VIII |
| 44 | 1999 | 25 | December | Kamashi | 38,64 | 66,42 | 5,1 | 12 | VII |
| 45 | 2000 | 21 | April | Kamashi | 38,68 | 66,52 | 5,0 | 10 | VII |
| 46 | 2000 | 19 | January | Kamashi | 38,66 | 66,5 | 5,0 | 10 | VII |
| 47 | 2007 | 27 | January | Sumsar | 41,38 | 71,31 | 5,1 | 12 | VI-VII |
| 48 | 2008 | 1 | January | Gulchin | 40,32 | 72,97 | 6,0 | 20 | VIII |
| 49 | 2008 | 28 | October | Jalal-Abad | 40,98 | 73,16 | 5,1 | 9 | VII |
| 50 | 2008 | 22 | August | Tashkent | 41,3 | 69,4 | 5,0 | 10 | VI-VII |
| 51 | 2011 | 19 | July | Kanskoe | 40,16 | 71,42 | 6,1 | 10 | VIII |
| 52 | 2013 | 24 | May | Tuyabogoz | 40,89 | 69,15 | 5,6 | 18 | VII |
| 53 | 2013 | 26 | May | Marzhanbulak | 39,96 | 67,34 | 6,1 | 18 | VIII |

| 54 | 2017 | 29 | September | Bakhmal | 39,75 | 67,91 | 5,1 | 5 | VI-VII |

Note: Earthquakes marked with an asterisk (*) are historical.

Risk assessment is crucial for preventing major disasters in the event of a significant seismic threat. The first systematic studies on seismic risk assessment, conducted about 60 years ago, laid the groundwork for future activities (Cornell, 1968; Algermissen et al., 1972; Keilis-Borok et al., 1973; Whitman et al., 1975; Lomnitz and Rosenblueth, 1976). In recent decades, particularly during the International Decade for Natural Disaster Reduction (IDNDR, 1990–2000), the global community has increasingly recognized the significance of the issue. The shift in focus from hazard to risk, driven by a series of devastating earthquakes worldwide, has prompted the development of procedures and techniques for assessing seismic vulnerability, damage, and conducting risk analysis on various geographical scales, e.g., PELEM (1989), Chen et al. (1992, 2002), Papadopoulos and Arvanitides (1996), King et al. (1997), McCormack and Rad (1997), Zonno et al. (1998), FEMA-NIBS (1999), Faccioli and Pessina (2000), RADIUS (2000), Bendimerad (2001), Fah et al. (2001), Coburn and Spence (2002), Lang (2002), Frolova et al. (2003), Giovinazzi and Lagomarsino (2004), Mouroux et al. (2004), Schwarz et al. (2004), Trendafiloski and Milutinovic (2004), Tyagunov et al. (2006), Di Pasquale et al. (2005), Wang et al. (2005), Alkaz et al. (2012) and many others. Different interpretations of the risk concept can be found in different publications, although the general consensus is that risk is a quantified possibility of losses. In the study by Erdik et al. (2004), the seismic risk of the cities of Tashkent and Bishkek was assessed using a scenario earthquake. Tyagunov et al. (2012) evaluated the seismic risk of Central Asian countries. The combined aspects of the seismic hazard distribution, seismic vulnerability and exposed assets provide the necessary basis for seismic risk analysis. A similar analysis of the territory of Uzbekistan was the goal of this study, conducted as part of the implementation of the above paragraphs of the Decree of the President of the Republic of Uzbekistan dated July 30, 2020, No.4794, by the Institute of Seismology of the Academy of Sciences of the Republic of Uzbekistan.

To develop a seismic risk map for the territory of Uzbekistan, databases were created based on GIS platforms allowing systematization and evaluation of the regional distribution of information on seismic hazard, number of buildings and structural types, geographical location of residential buildings, coefficient of the seismic vulnerability of buildings and territories, cadastral building value, etc.

Seismic vulnerability analysis was conducted using GESI_Program, which is based on the methodology for the assessment of seismic damage to buildings. At the same time, the existing buildings in the territory of the republic were collected and classified according to the building structural type. There are 5 types of buildings: buildings built using local clay materials, brick buildings, wooden buildings, buildings constructed using a metal frame and reinforced concrete buildings. In previous studies of urban and regional territories, seismic data analysis considered the influence of local soil conditions (microzoning and detailed zoning), inventory of buildings and asset values (element-by-element inventories or based on representative units) (Ismailov et al. (2022a), Ismailov et al. (2022b), Ismailov et al. (2023a)).

The developed seismic risk map of the territory of the Republic of Uzbekistan was based on an assessment of probable economic losses within administrative districts combined with seismic hazard factors, seismic vulnerability and concentration of values, ranging from zero to hundreds of trillions of Uzbekistan soms. It is important to emphasize that the level of seismic hazard used in the calculation of physical and economic damage corresponds to a 90% probability of not exceeding seismic impacts over a period of 50 years, which corresponds to an average return period of 475 years. This level of probability is the generally accepted standard in seismic hazard assessment during the design and construction of conventional buildings and structures. Of course, considering a different probability, level of hazard and consequently, the assessment of damage and potential losses may differ from the data presented.

In the development of map of seismic risk for the territory of the Republic of Uzbekistan, seismological and macroseismic databases of the Institute of Seismology of the Academy of Sciences of the Republic of Uzbekistan[2], database on the housing stock of Uzbekistan of the State Cadastral Agency under the Tax Committee of Uzbekistan[3] and research experience and publications of the Institute of Seismology[1] and the Institute of Mechanics and Seismic Stability of Structures of the Academy of Sciences of the Republic of Uzbekistan[4] and JSC ToshuyjoyLITI[5] were considered during the implementation of this research.

The present study is concentrated on the assessment of direct economic losses that may be caused by structural damage to residential buildings as a result of seismic actions. At the same time, given that residential buildings predominate in the development of cities and administrative districts in Uzbekistan, the presented results could serve as a clear reference for a comparative analysis of seismic risk in various administrative districts.

## 2. Data and methods

### 2.1. Characteristics of the engineering-geological conditions

Based on the analysis of geomorphological and geologic-lithological structure, as well as groundwater distribution and exogenous geological processes, engineering-geological zoning has been conducted.

The peculiarities of the engineering-geological conditions of Uzbekistan's territory have been identified and described in the works of Mavlyanov et al. (1987), Kasymov (1979), Ismailov et al. (1968) and others. The engineering-geological map of the

[2] https://seismos.uz/

[3] https://kadastr.uz/uz

[4] https://instmech.academy.uz/ru

[5] https://toshuyjoyliti.uz/

Republic of Uzbekistan is divided by lithologic composition into 14 districts (Rock soils; Limestones; Sands and sandstones; Clays and sands; Clays, marls, sandstones; Clays, sandy clays, sands; Gravels; Sands; Sands, sandy clays, sands; Sandy loams, sands; Gravels, pebbles, rubbles; Loess soils, loess loams and sandy clays; Clays, loams and sandy clays; Gypsum, loams and clays) (Fig. 1).

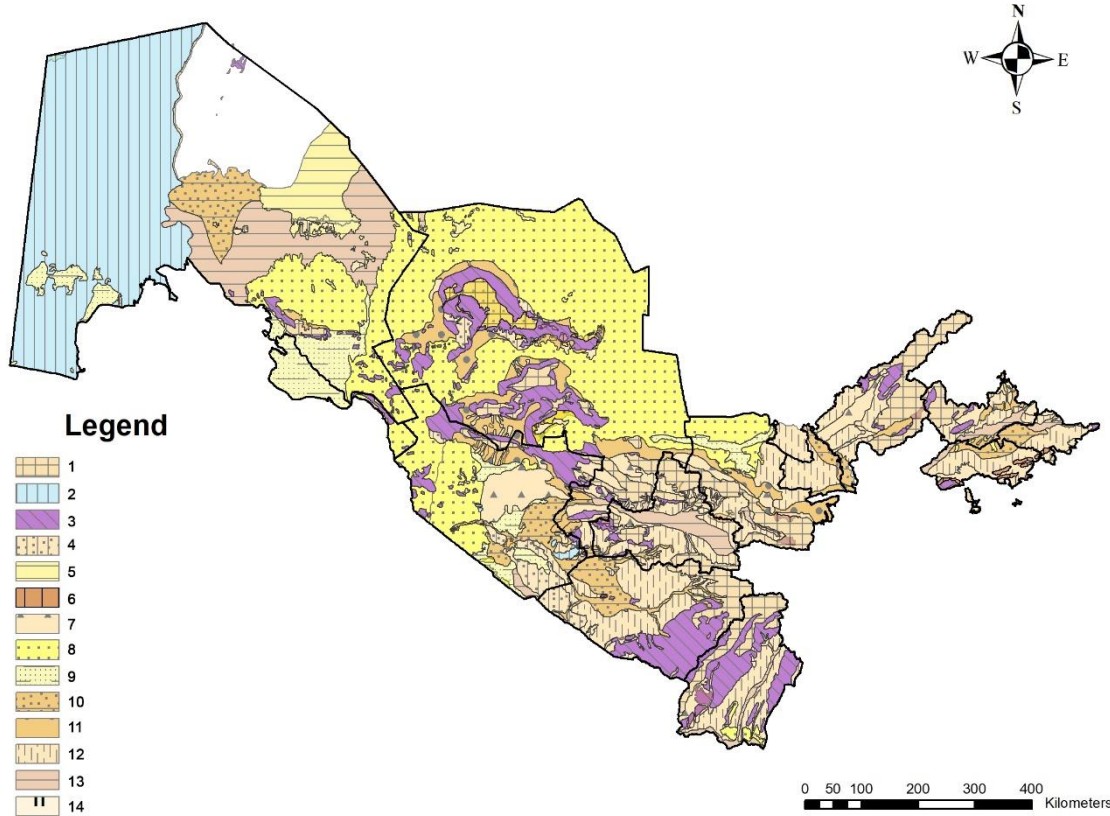

**Fig. 1:** Engineering-geological map of Republic of Uzbekistan. Autors: Ismailov et al. (1968). 1-Rock soil; 2-Limestone; 3-Sand and sandstone; 4-Clay and sand; 5-Clay, marl, sandstone; 6-Clay, sandy clay, sand; 7-Gravel; 8-Sand; 9-Sand, sandy clay, sand; 10-Sandy loam, sand; 11-Gravel, pebble, rubble; 12-Loess, loess loam, and sandy clay; 13-Clay, loam and sandy clay; 14- Gypsum, loam and clay.

The complexity of geological structure of the upper soil layers (10-15 m), the diversity of petrographic and lithological composition of soils, the geomorphological characteristics and the unique climate determine the variety of engineering-geological conditions in Uzbekistan's territory. The main features of the republic's orography are closely related to the peculiarities of the geological structure of numerous mountain ranges. Wide plains, intermountain uplifts and depressions are located between the mountain ranges, characterized by an abundance of weathering products Kasymov S.M. (1979).

The complexity and diversity of the engineering-geological conditions in Uzbekistan can be explained by the broad distribution of different geological and lithological strata, which exhibit a certain zoning. While metamorphic, igneous, and sedimentary rocks are developed in mountainous and foothill areas, gravel, pebbles, sands and loamy deposits are prevalent in the vicinity of mountains. Aeolian and alluvial loams, loess soils and sands are widespread in lowland areas.

       The first groundwater tables are distributed at various depths depending on the geomorphological structure. The highest
groundwater levels are observed in the plains, especially in areas with active agricultural land development. Groundwaters in rock deposits is mainly confined to fractures and fault zones.

       Exogenous geological processes are primarily developed in mountainous and foothill plains and are represented by landslides, rockfalls and soil erosion.

       The seismic risk probability and economic map of the administrative districts of the Republic of Uzbekistan were developed
based on the engineering geological conditions and general seismic zoning maps. Subsequently, seismic vulnerability levels were assessed using the GESI_Program software developed by the RADIUS program of the International Federation of Red Cross and Red Crescent Societies during 1999-2001. The assessment considered various construction materials based on cadastral information, considering the types of buildings and their vulnerability functions. The seismic vulnerability levels of buildings were then evaluated in the districts of the Republic. Considering the ground conditions, the economic map of seismic
risk probability in the administrative districts of Uzbekistan was developed, showing the probability of not exceeding 50% within 90 years (in trillion soums).

**2.2. Assessment of seismic hazard considering soil conditions and comparison with previous studies**

Variation of seismic intensity increments across the territory of Uzbekistan has been examined. An improved map of seismic zoning of the territory of the Republic of Uzbekistan (Artikov et al. (2020) (OSR-2017) has been compiled, considering the seismic properties of soils of different categories (Fig. 2).

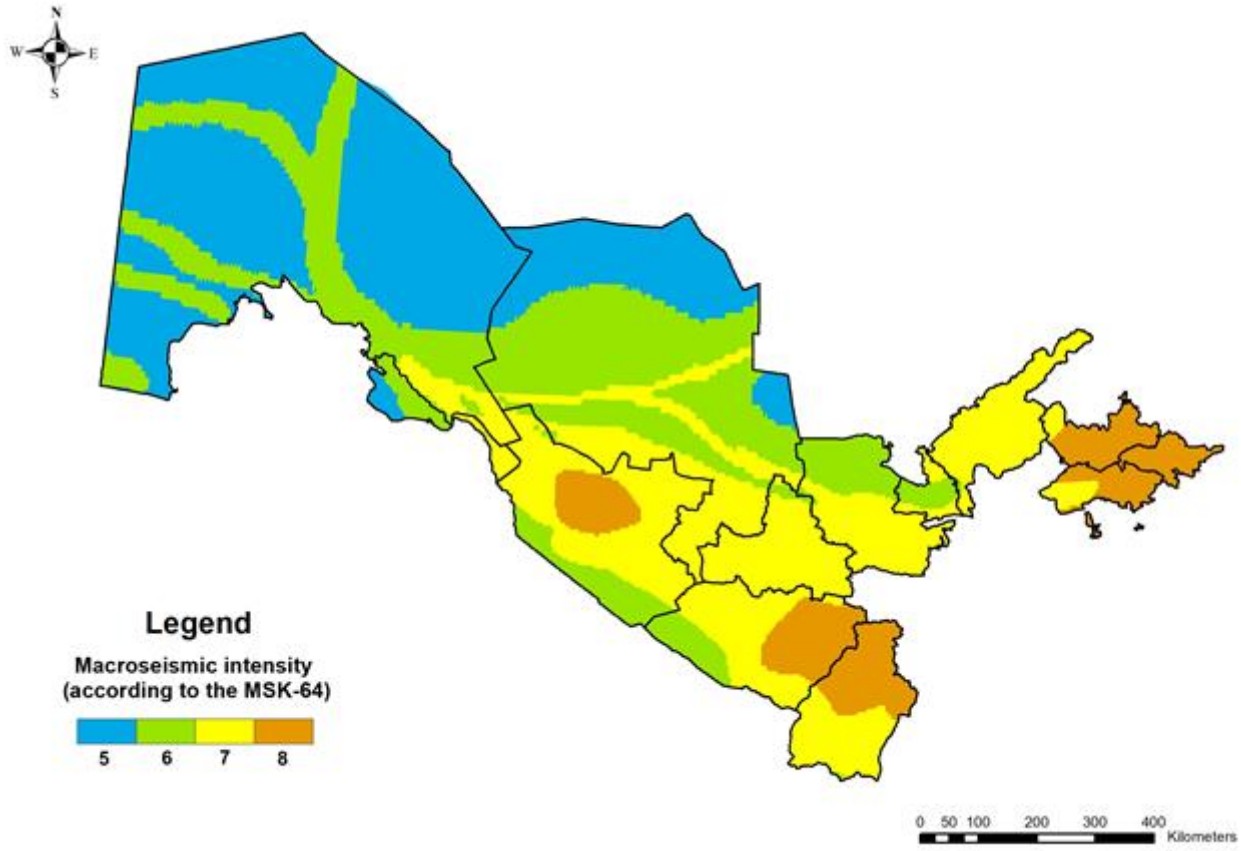

**Fig. 2:** Map of seismic zoning of the Republic of Uzbekistan (OSR-2017).
Map of seismic zoning of the Republic of Uzbekistan (probability of not exceeding P=90% in 50 years)
Authors:T.Artikov, R.Ibragimov, T.Ibragimova, M.Mirzaev

 In the National Building Code No.2.01.03-19 "Construction in Seismic Areas", soils have been systematically classified into three categories based on their seismic properties, with corresponding seismic intensity increments established for each
category, taking into account the engineering-geological conditions of the soils. The assessment specifically targeted the upper 10-meter strata. For the 1st category, encompassing rock soils, the seismic intensity increment is reduced by 1. This adjustment is based on the observation that structures within the region tend to experience a lower intensity, typically differing by approximately -1 from the regional intensity during an earthquake. Similarly, the 2nd category, comprising sandy and analogous soils, maintains the same seismic intensity as the considered region. In contrast, the 3rd category, encompassing
clays, loess, and other soils with limited seismic resistance, witnesses a seismic intensity increment increased by 1. The general seismic zoning OSR-2017 (Fig. 2) is calculated based on the 2nd category of soils. Using Fig. 1 of the engineering-geological conditions of the territory of the Republic of Uzbekistan and the general seismic zoning OSR-2017 (Fig. 2), we compiled the schematic map of seismic intensities in the territory of the Republic of Uzbekistan (Fig. 3).
 Moreover, "Regional seismic risk assessment based on ground conditions in Uzbekistan" was a pilot project covering the entire
territory of Uzbekistan. We have assessed seismic risk for the Djizak region and the city of Tashkent. Based on geological, seismotectonic, and seismological conditions, a scenario earthquake has been identified for the seismic risk assessment of the Djizak region and the city of Tashkent (RADIUS). Moreover, the social (individual) seismic risk for the Andijan region was calculated using the scenario earthquake.
 In the RADIUS (1992) project, the seismic risk of the city of Tashkent was assessed using a scenario earthquake. The total
damage from the scenario earthquake, considering the destruction of life support systems and infrastructure in Tashkent, is estimated at about 1 billion Uzbekistani soms. (The loss figures are determined in prices for the 1991 period and are taken at the book value, significantly underestimated.) As Tashkent is the capital, where a quarter of the country's gross domestic product is produced, the consequences of an earthquake will undoubtedly affect the entire country. Many international commercial, banking, and insurance connections will be temporarily disrupted. Human casualties will be significant. Years
will be needed for the recovery of economic losses. In addition, the shutdown of industrial production is expected to result in losses of about 1 billion U.S. dollars. Preliminary calculations show that the scenario earthquake will cause damage to the city totaling more than 10 billion U.S. dollars (taking into account the book value of fixed assets determined at 1991 prices). Expert estimates suggest that about 80% of communication facilities will be out of operation for an extended period. Ongoing construction projects will incur irreparable damage amounting to approximately 1 billion U.S. dollars.

To assess individual (social) seismic risk, a map of a scenario earthquake was created using the GIS "Extremum," developed by the Center for Emergency Situations in collaboration with the Seismological Center of the Institute of Geoecology of the Russian Academy of Sciences and the Scientific Research Institute of the State Ministry of Civil Defense, Emergencies and Elimination of Consequences of Natural Disasters of Russia. Data from the Andijan earthquake of 1902 were used. Based on calculations, a map of the individual seismic risk of the Andijan region and adjacent areas was constructed. It is estimated that
the loss of population could amount to 8,260 people, and the total losses (including injuries) could reach 13,440 people.

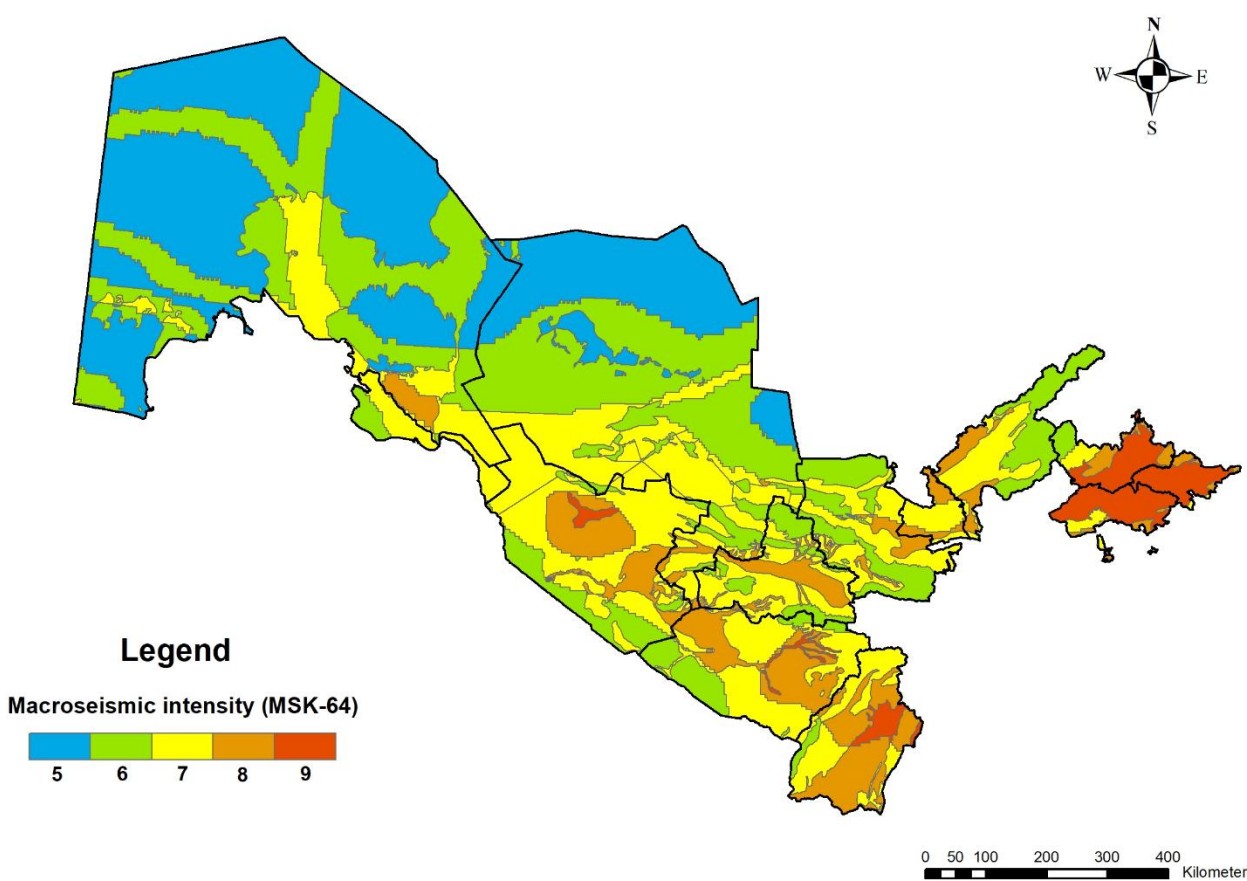

**Fig. 3.** The schematic map of seismic intensity in the territory of the Republic of Uzbekistan.

The assessment of seismic risk in the territory of the Republic of Uzbekistan was conducted taking into account the experience
of such countries as Germany (Tyagunov et al., 2006), Italy (Pasquale et al., 2005) and Russia (Zaalishvili et al., 2019). The basis for this assessment was the seismic zoning map for a 90% probability of not exceeding seismic effects over a 50-year period. Therefore, we utilized the seismic zoning map of the territory of Uzbekistan (OSR-2017) (Artikov et al., 2020) for evaluation of seismic hazard of the territory.
In accordance with the local building code[6], all soils have been systematically classified into three categories based on their
seismic properties, and corresponding seismic intensity increments have been determined for each category. The evaluation focused on the upper 10-meter strata. For the 1st category, encompassing rock soils, the seismic intensity increment was reduced by 1. This adjustment is rooted in the observation that when the region is subjected to an earthquake, structures within it experience a lower intensity, typically differing by approximately -1 from the regional intensity. Similarly, the 2nd category, consisting of sandy and analogous soils, maintains the same seismic intensity as the considered region. In contrast, the 3rd
category, which includes clays, loess, and other soils with limited seismic resistance, witnesses a seismic intensity increment increased by 1. Consequently, a seismic intensity increment map has been compiled at a scale of 1:1000000. The Republic of Uzbekistan has been partitioned into zones reflecting seismic intensity increments of -1, 0, and 1. In simpler terms, this map delineates areas where the same earthquake may induce more significant destruction due to unfavorable soil conditions and areas where the impact would be comparatively reduced.
Based on the compiled map of seismic intensity increments, adjustments have been made to the OSR-2017 map (Artikov et al., 2020)  (Fig. 2). As a result, the seismic intensity for the entire territory of Uzbekistan has been determined, taking into account the soil categories based on seismic properties. Fig. 3 shows the map of seismic intensity in macroseismic units developed using a methodology (Fig. 4) that incorporates soil conditions in assessing earthquake intensity. As can be seen on the map (Fig. 3), a zone with an intensity of 9 has appeared, which indicates that there were unfavorable soil conditions such
as areas with clays or loess soils with high level of water table (Table 2).

---

[6] Building code of the Republic of Uzbekistan No.2.01.03.19

**Table 2.** Comparison of the ratio of areas with different intensities (based on the MSK-64 macroseismic scale) between two seismic hazard maps, one considering ground conditions and the other not

| | 5 | 6 | 7 | 8 | 9 |
|---|---|---|---|---|---|
| Seismic hazard map | 31,1% | 26,8% | 31,8% | 9,3% | |
| Seismic hazard map with consideration of ground conditions | 16,2% | 39,5% | 27,1% | 10,7% | 6,5% |

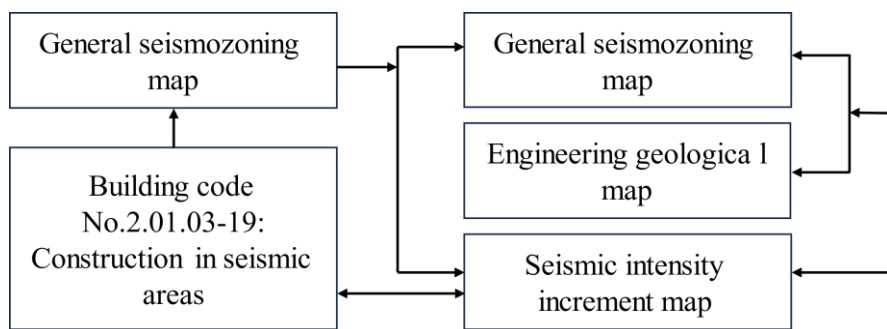

**Fig. 4:** Procedure to compile the seismic hazard map of the Republic of Uzbekistan

There are some differences in values and established boundaries of seismic hazard zones between the OSR-2017 map and the compiled map of seismic intensity in the territory of the Republic of Uzbekistan. These differences are due to the delineation of different zones based on seismic intensity parameters, related to the distribution of soils of Categories I and III. For example, in the OSR-2017 map, the zone with I=7 is subdivided into zones with intensities of 6, 7, and 8 on the seismic intensity map, depending on the soil conditions. However, the entire territory of the republic is divided into zones with seismic intensities of 5, 6, 7, 8, and 9.

### 2.3. Seismic vulnerability

Seismic vulnerability of buildings is to the ratio of expected costs of restoring structures that may be subjected to destructive seismic events of a given intensity, to their initial cost. Vulnerability ranges from 0 (no damage) to 1.0 (unrepairable). By knowing the current value of a structure, the monetary damage can be determined. The relationship between vulnerability and seismic impact (e.g., in degrees) is referred to as the vulnerability function. Vulnerability functions play a central role in regional seismic loss assessment.

A vulnerability function represents the relationship used to forecast statistics (such as mean value or standard deviation) of seismic losses distribution. It predicts the extent of damage that a structure (e.g. residential building or bridge) will experience under probability of seismic events. It should be noted that vulnerability functions are calculated separately for each type of building listed in the cadaster.

Vulnerability functions for the identified structural building types within the territory of the Republic of Uzbekistan were developed using the "GESI_Program", which is a computer program based on the assessment of structural damage under specified seismic events (Fig. 5), which we used for the vulnerability of buildings to assess the seismic risk of the territory of the Republic of Uzbekistan. This software was developed as part of the United Nations' Global Earthquake Safety Initiative (GESI) Pilot Project in 1999-2001. The primary data used for the program's development was collected within the framework of the international RADIUS project (Risk Assessment Tools for Diagnosis of Urban Areas against Seismic Disasters), conducted by the UN-IDNDR Secretariat in 1998-1999. The vulnerability function used to assess seismic risk was created in an experiment involving cities such as Addis Ababa (Ethiopia), Antofagasta (Chile), Bandung (Indonesia), Guayaquil (Ecuador), Zigong (China), Izmir (Turkey), Skopje (Macedonia), Tashkent (Uzbekistan) and Tijuana (Mexico). The experiment utilized identical building materials in the respective cities. The vulnerability index for the city of Tashkent in the experiment did not exceed 10% of the total (RADIUS, 2000).

In Fig. 5, in addition to the vulnerability functions, the boundary conditions of damage are also presented, characterized by the overall direct costs of restoring buildings to their initial condition and the relationship between PGA and intensity (according to the MSK-64 macroseismic scale) was calculated using the equation $I_{max}=0.41I-0.755\pm0.08$ (Aptikaev, 2012).

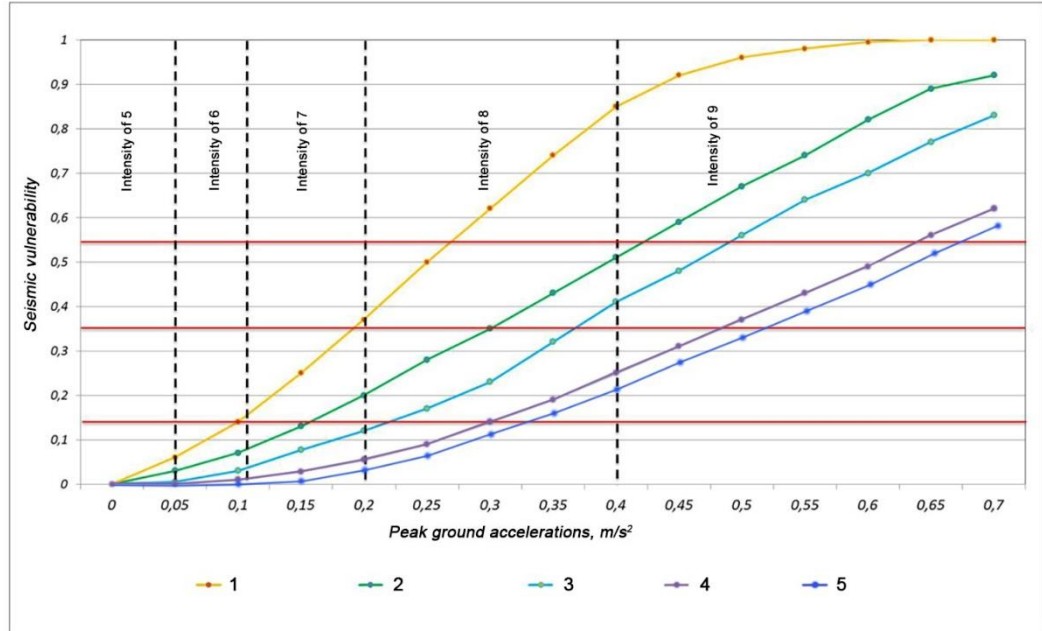

**Fig. 5:** Vulnerability function for the different building types. 1-Adobe (local); 2- Masonry; 3- Wooden; 4- Concrete; 5-Metal frame.

Buildings built using local materials (adobe, guvalyak, pakhsa and raw bricks); masonry buildings; wooden (chopped or panel) residential buildings; concrete (panel, large-panel, monolithic and reinforced concrete) buildings; and buildings with a metal frame or a frame with diaphragms (ties).

As of February 1, 2021, at the republican level, 7,135,881 residential buildings were analyzed and systematized by employees of the Institute of Seismology of Academy of Sciences of Uzbekistan with a total area of 4.4 billion square meters. These buildings were categorized by the material of structural system and aggregated by administrative regions (Table 3).

**Table 3**. Residential buildings by the material of structural system within zones of different seismic intensities

| Seismic intensity zones | Total | Residential buildings by structural types | | | | |
|---|---|---|---|---|---|---|
| | | RC | Wooden | Masonry | Metal frame | Local adobe materials |
| 5 | 6031 | 758 | 1 | 2933 | 0 | 2339 |
| 6 | 398838 | 24431 | 3323 | 62787 | 126 | 308171 |
| 7 | 1956323 | 176113 | 10029 | 338873 | 3292 | 1428018 |
| 8 | 2960146 | 169079 | 31954 | 985165 | 6318 | 1767630 |
| 9 | 1819597 | 133535 | 23030 | 217787 | 3025 | 1442220 |

The vulnerability function for each structural type of buildings was determined using the GESI_Program, which served as the basis for calculating seismic vulnerability by administrative regions. For aggregation of values of seismic vulnerabilities of buildings, the equation proposed by Tyagunov, S.A. et al. (2007) was used.

$$MRV = \frac{\sum_{i=1}^{n} N_i \cdot MVR_i}{\sum_{i=1}^{n} N_i}$$

Here, MVR represents the average value of seismic vulnerability for the territory of the district, $MVR_i$ represents the average value of seismic vulnerability for the identified structural types of buildings and N represents the number of buildings by structural types within the administrative district.

Thus, administrative districts with seismic vulnerability values of 0-0.15, 0.16-0.3, 0.31-0.45, 0.46-0.6 and 0.61-0.75 were identified. These values were aggregated to create a schematic map of seismic vulnerability for the administrative districts of the Republic of Uzbekistan (see Fig. 6).

The schematic map of seismic vulnerability is the basis for the assessment of possible damage at given values of seismic impacts.

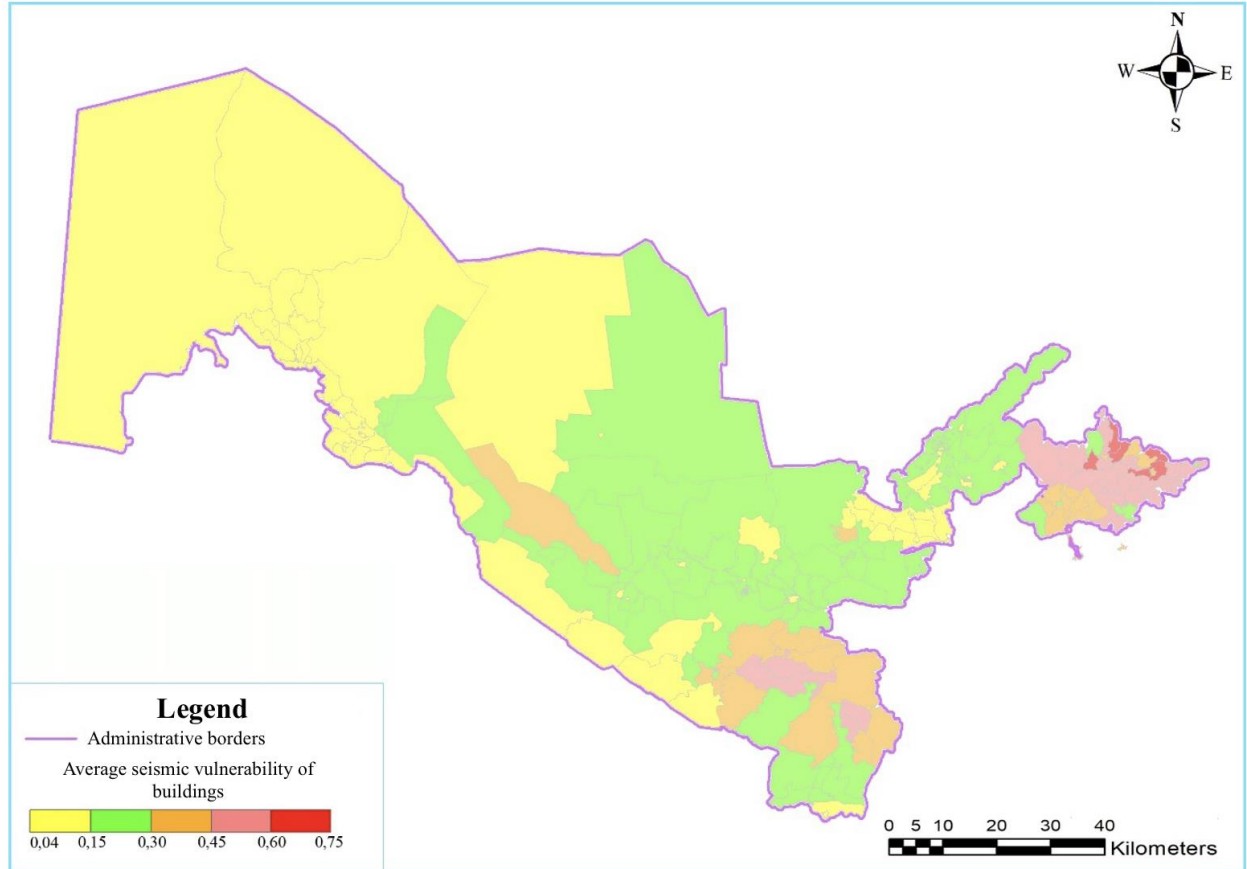

**Fig. 6:** Map of average values of seismic vulnerability of buildings by administrative districts of the Republic of Uzbekistan

The GESI_Program[7] consists of five sets of input parameters that characterize the type of structure, design features, quality of construction, quality of building materials and level of seismic impacts in the peak acceleration process. Based on these input parameters, a damage diagram and building vulnerability function are constructed. Damage to buildings is rated at four levels: light, moderate, heavy and very heavy (table 4).

**Table 4.** Damage characteristics of buildings:

| Grade | State of damage | Description |
|---|---|---|
| 1 | Minor | - light non-structural damage, including cracks in plaster (up to 0.5 mm wide), chipping of small plasters from walls and frame elements, and thin cracks in partitions, cornices and floor screeds.<br>- light structural damage (complete or almost complete absence). Minor damage requires maintenance costs. According to norms, the cost can reach up to 15% of the book value of the object. |
| 2 | Moderate | - moderate non-structural damage, including chipping of rather large pieces of plaster, falling roof tiles, cracks in chimneys, falling parts of chimneys, through cracks in partitions and lintels above openings, cracks in the masonry of gables and parapets, and their partial displacement.<br>- light structural damage, including small cracks in walls, between prefabricated floor panels, along the counter of large blocks, and in the load-bearing elements of frames. Overhaul costs are calculated based on damage to the building, ranging from 15–35%. |
| 3 | Severe | - severe non-structural damage, including falling chimneys, gable wall parapets, collapse of individual or many load-bearing and self-supporting elements, and destruction of lintels over openings.<br>- moderate structural damage, including large deep and through cracks in walls, loss of connections between structural elements and separation of longitudinal walls from transverse ones. In the case of severe damage, the restoration costs are determined depending on the nature of the damage and are decided by an expert commission. Restorative repair is determined depending on the damage, ranging from 35% to 55%. |

---

[7] https://iisee.kenken.go.jp/net/saito/gesi_program/index.html (retrieved on September 21, 2023)

| 4 | Very severe | - non-structural destruction, including the collapse of individual sections of internal walls and collapse of partitions.<br>- structural destruction, including delamination of the masonry of load-bearing walls, gaps in walls, destruction of connections between individual parts of the building, and rupture of the joints of prefabricated structures. In case of damage to the building of the 4th degree, the building is subject to demolition |
|---|---|---|

The seismic vulnerability is estimated as a percentage of the damage due to peak acceleration.

According to the definition, the vulnerability of buildings is considered a property of a given structure capturing the loss of qualitative or quantitative indicators of reliability and safety due to any impact. The vulnerability ranges from 0 (no damage) to 1 (unrepairable). The dependence of the vulnerability on seismic impact (for example, in intensity) is denoted as the vulnerability function.

The vulnerability function relating the degree of damage to the level of seismic impact, given in intensity is usually determined empirically.

For a detailed assessment of the damage to buildings under different intensities of seismic impacts and to compile vulnerability functions for specific structural types of buildings, calculations were performed in the GESI_Program.

A comparison of the results revealed (Fig. 7) that macroseismic observations of the damage to the buildings under consideration
greatly differ from the calculation results obtained with GESI_Program, but at the intensity of 7.3 the observations and calculation results coincide. Closer matches are shown in graphs obtained via calculation using GESI_Program (RADIUS, 2000) and experimental data of Khakimov Sh. (2017). Based on these data, it can be assumed that the use of GESI_Program in the assessment of the vulnerability of various construction types of buildings yields better results, at least excluding subjective opinions when comparing the vulnerability of buildings.

The vulnerability function, which relates the degree of damage to the level of seismic impact, given in MSK-64 intensity or peak ground acceleration values, is usually determined empirically or via calculation methods. When studying the engineering consequences of strong local earthquakes, world statistics of damage data for classes of objects located in the study area under similar seismogeological conditions are involved. To date, the Institute of Seismology of the Academy of Sciences of Uzbekistan has accumulated a large amount of data on the consequences of strong earthquakes. However, the range of observed
intensities remains insufficient to obtain full-fledged regional loss matrices. Therefore, at this stage, we limited ourselves to using the GESI_Program, which, at the moment, is the best way to model and evaluate the relationship between the degree of damage and level of seismic impact.

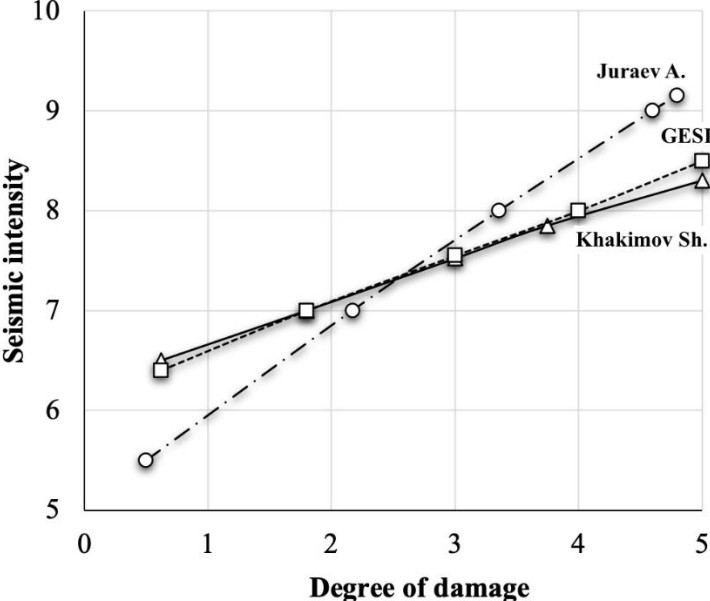

**Fig. 7:** Graph of changes in the average degree of damage to individual mudbrick houses depending on the seismic intensity
according to different authors.

This paper represents the first attempt to compile an extensive database of residential buildings in Uzbekistan and involves significant efforts to include the most at-risk assets in the territory. At the same time, a database of the housing stock in the republic was formed based on the database of the State Cadastral Chamber of the Cadastral Agency under the Tax Committee of Uzbekistan.
Residential buildings in the territory of Uzbekistan could be divided into 5 main types of structural systems:

       1. Type A: local adobe materials (guvalyak, pakhsa and raw bricks);
       2. Type B: masonry.
       3. Type C: wooden (chopped or panel);
       4. Type D: concrete (panel, monolithic and reinforced concrete);
300        5. Type E: metal frame or a frame with diaphragms (ties).

These 5 types of buildings could be subdivided into 24 different subtypes according to their structural features and year of construction (Table 5). This classification of the buildings is typical not only for Tashkent, but also for other cities in Central Asia. The buildings were also classified according to the number of stories and type of material of the supporting structures.

**Table 5.** Classification of buildings in Tashkent according to the vulnerability index (Khakimov Sh., 2000)

| No. | Building types and their structural types | Average damage index |
|---|---|---|
| 1 | Residential buildings constructed from local low-strength materials (without anti-seismic measures) | 3.95 |
| 2 | One-story clay walls of the guvalyak and pakhsa types | 3.68 |
| 3 | Three- to five-storey frameless brick buildings with wooden floors constructed until 1958 | 3.84 |
| 4 | Prefabricated reinforced concrete frame made of linear elements with a welded joint in the zone of maximum effort, or the same with stiffening diaphragms in one direction (framework III of the IIS-04 series and their modifications) | 2.96 |
| 5 | One- to two-storey frameless brick walls with wooden floors | 3.15 |
| 6 | Crossbarless frames or buildings erected by raising floors (crossbarless frame with stiffening core) | 2.75 |
| 7 | Buildings with a flexible ground floor and rigid upper floors | 2.7 |
| 8 | Walls made of bricks, small concrete or natural stones; ceilings - prefabricated reinforced concrete | 2.62 |
| 9 | Large-panel walls without anti-seismic measures | 2.61 |
| 10 | Buildings with external load-bearing brick walls; internal - reinforced concrete frame elements | 2.58 |
| 11 | Prefabricated frame of flat reinforced concrete cross or H-shaped elements with monolithic nodes | 2.56 |
| 12 | Monolithic reinforced concrete frame | 2.55 |
| 13 | Walls made of large blocks (concrete, vibro-brick, or reinforced vibro-brick panels) | 2.5 |
| 14 | Reinforced concrete frame with brick filling | 2.41 |
| 15 | One- to two-storey wooden frames filled with raw bricks (sinch) | 2.37 |
| 16 | Walls of complex construction (with reinforced concrete inclusions); ceilings - prefabricated reinforced concrete | 2.33 |
| 17 | Prefabricated reinforced concrete frame-braced frame with monolithic nodes, with stiffening diaphragms in two directions or stiffening cores | 2.22 |
| 18 | Frame made of spatial elements (volumetric cross) with monolithic knots | 2.17 |
| 19 | Large-panel buildings with brick exterior walls | 2 |
| 20 | Monolithic walls | 1.86 |
| 21 | Large panel walls | 1.73 |
| 22 | Volumetric blocks per room | 1.67 |
| 23 | One- to two-storey wooden houses (chopped or panel) | 1.16 |
| 24 | Metal frame or frame with diaphragms (bonds) | 1.16 |

Notes:
1. The table provides average values of the damage index.
2. The first column of the table indicates the degree of vulnerability in the ascending order, the most vulnerable to the least vulnerable structural types.

We have taken the classification data of buildings from the database of the cadaster agency of Uzbekistan. For reference, the comparison between our data and EMCA is presented in Table 6.

**Table 6.** Classification of buildings in Tashkent according to the vulnerability index

| | Our classification | EMCA | | |
|---|---|---|---|---|
| | | EMCA Classification | Subtype | Description |
| 1 | Adobe (local) | EMCA4 | ADO | Adobe structures |
| 2 | Masonry | EMCA1 | CM | Brick masonry of a complex structure |
| 3 | Wooden | EMCA5 | WOOD1 | Wooden structure, load-bearing frames with connections |
| | | | WOOD2 | Wooden structure, wooden frame, and adobe infill |

| 4 | Concrete | EMCA2 | All subtypes EMCA2 | All descriptions of EMCA2 subtypes |
| | | EMCA3 | All subtypes EMCA3 | All descriptions of EMCA3 subtypes |
| 5 | Metal frame | EMCA6 | STEEL | Steel structures |

**2.4. Distribution of residential buildings by the material and their cadastral value**

According to cadastral data, as of February 1, 2021, the housing stock in Uzbekistan consists of 7135881 houses and apartments.

Depending on the demographic situation, the number of residential buildings in the territory of the republic exhibits a very uneven distribution. The housing stock in Uzbekistan is divided in 2 main types: individual houses (80.1%) and multi-story residential buildings (19.9%). Individual houses are typically one or two-story buildings intended for one or two families, while residential buildings consist of multiple separate apartments.

It should be noted that in the housing stock of the republic, there are 44827 multi-story buildings, where there are 1375623 apartments, which are also considered when compiling the residential building database.

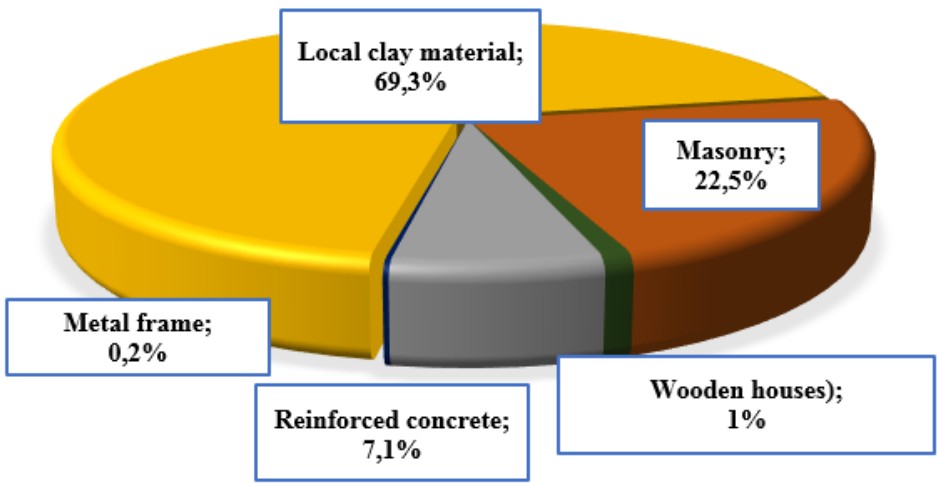

**Fig. 8:** Distribution of residential buildings by the material of structural system.

These types of buildings are distributed unevenly in quantitative terms and spatially, so among these buildings, buildings built using local materials are the most widespread. These buildings are highly represented in rural areas (settlements, towns, cities, etc.) and comprise about 70% of the total number of residential buildings in Uzbekistan. The buildings built of wood (including panel houses) or metal frames, comprise less than 1% of the total number of residential buildings. Figure 8 shows the distribution of residential buildings by the material of structural system.

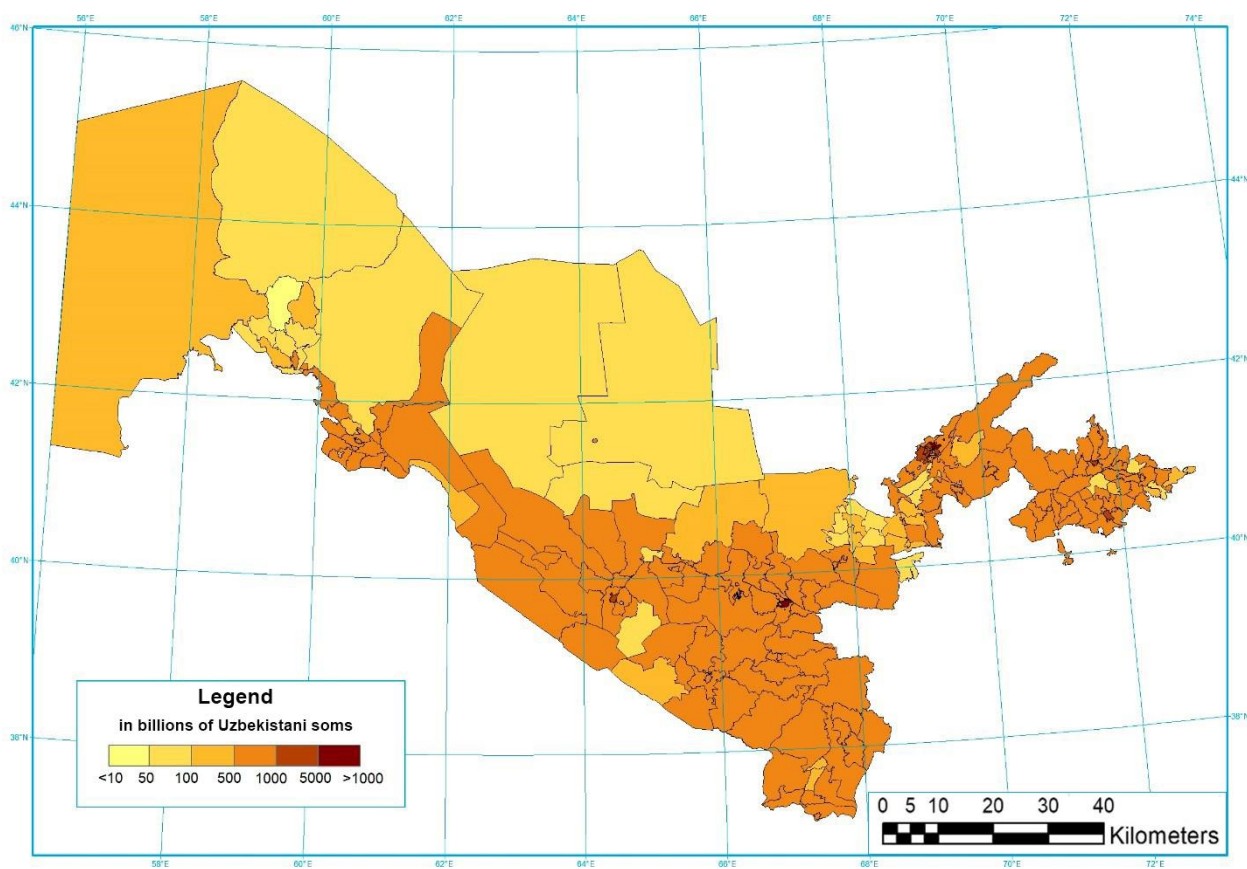

**Fig. 9:** Map of the total cadastral value of the housing stock within the administrative regions of the Republic of Uzbekistan.

Cadastral value of residential buildings by administrative areas is also an important information for developing maps of seismic risk, as well as for the government that implementing policies for increasing the seismic resilience of buildings and structures. Figure 9 shows the cadastral value of housing stock within the Republic of Uzbekistan and its administrative areas.

Based on given data, seismic risk assessment of the territory of the Republic of Uzbekistan will be performed in the next chapter.

**3. Seismic risk assessment**

Analysis of given data demonstrates a large spread in the number of buildings by structural types. For example, in the Kashkadarya region, the share of buildings built from local clay material exceeds 83% (27 trillion Uzbekistani soms) of the total number of residential buildings; in the Samarkand (40 trillion Uzbekistani soms) and Andijan regions (21 trillion
Uzbekistani soms), the share is 82%; and in the Tashkent region, 48.3% (16 trillion Uzbekistani soms). In large cities, the percentage of adobe residential buildings is smaller and ranges from 13% to 27%. This circumstance must be considered when assessing the seismic risk, since the amount of damage due to an earthquake in the selected territorial units depends on the proportion of the specific structural types of buildings.
The number of residential buildings located in the territory with different seismicity values, expressed by peak ground
accelerations is shown in Figure 10. This diagram shows that a large number of buildings, approximately 31% of the total number of residential buildings are located in the territory with PGA ranging from 100 to 150 cm/s$^2$, 27% of the buildings are located in areas with PGA of 0,15–0,20 m/s$^2$ and more than 30% are located in areas with peak accelerations higher than 0,20 m/s$^2$, representing the zone with an intensity of 8 (according to EMS-98, https://www.franceseisme.fr/EMS98_Original_english.pdf).

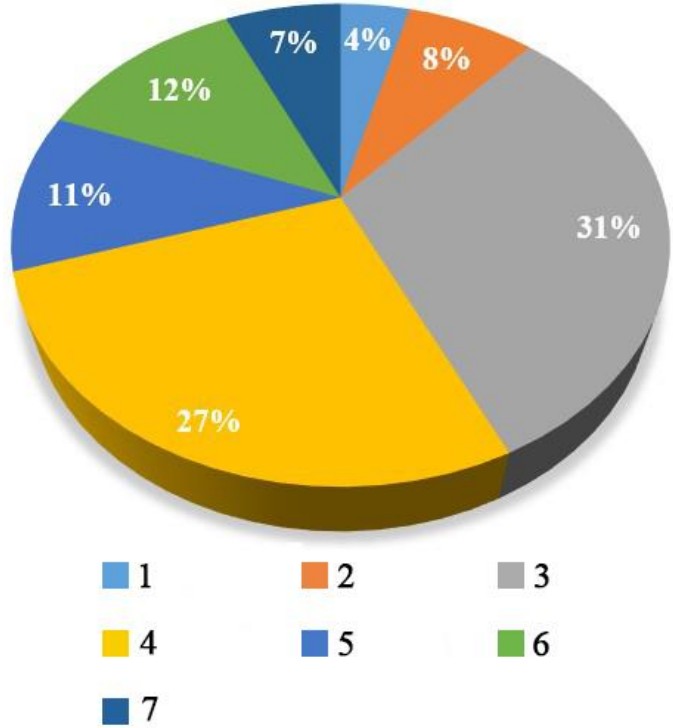

**Fig. 10.** Distribution of residential buildings in areas with different seismic effects (values of the peak ground acceleration are given in m/s²). **1:** 0-0.05; **2:** 0.05-0.10; **3:** 0.10-0.15; **4:** 0.15-0.20; **5:** 0.20-0.25; **6:** 0.25-0.30; **7:** 0.30-0.35

Information on the distribution of residential buildings by structural types depending on zones with different seismic effects is given in Tables 5 and 7.

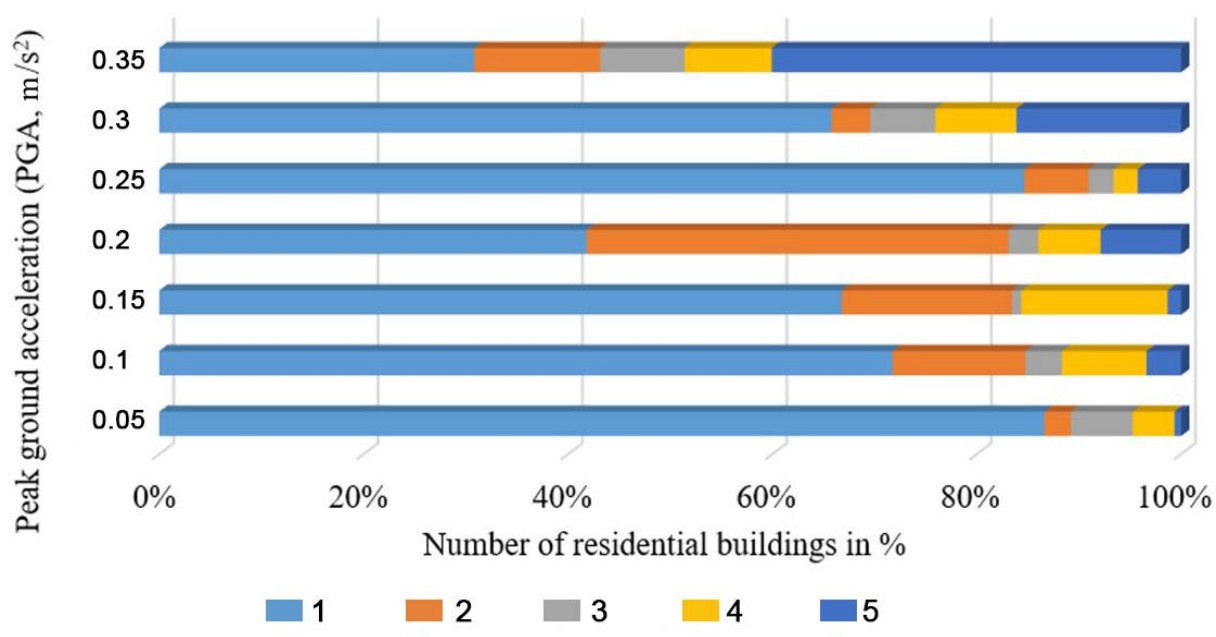

**Fig. 11.** Number of residential buildings by structural types located in the territory with different seismic effects (PGA, m/s²). **1:** Type A; **2:** Type B; **3:** Type C; **4:** Type D; **5:** Type E

**Table 7.** Distribution of residential buildings in Uzbekistan depending on the structural types of buildings (as of February 1, 2021)

| Structural type of the building | Total, % | including (%) | |
|---|---|---|---|
| | | in cities | in rural areas |
| Type A | 69,2 | 27,2 | 84,8 |
| Type B | 22,5 | 54 | 10,9 |
| Type C | 1 | 0,6 | 1,1 |

| | | | |
|---|---|---|---|
| Type D | 7,1 | 18 | 3,1 |
| Type E | 0,2 | 0,2 | 0,1 |

To assess the seismic risk within the context of the administrative districts of the Republic of Uzbekistan, it is necessary to take into account the share of the housing stock across all administrative districts, considering zones with different intensities. Figure 11 shows the percentage of residential buildings in Uzbekistan with different peak ground accelerations (compiled based on the OSR-2017 map with a probability of not exceeding 90%).

The spatial distribution of buildings within each administrative region significantly varies based on the seismicity of the respective territories. Figure 12 shows the distribution of residential buildings by regions in Uzbekistan and seismicity of areas where these buildings are situated (compiled based on the OSR-2017 map with a probability of not exceeding 90%). It can be seen that the central and left part of the country exhibit relatively lower seismicity, whereas areas with active faults, particularly in the western and southern parts of the city, pose a high risk to buildings. Many buildings in these regions are located in areas with elevated seismic activity and high seismicity.

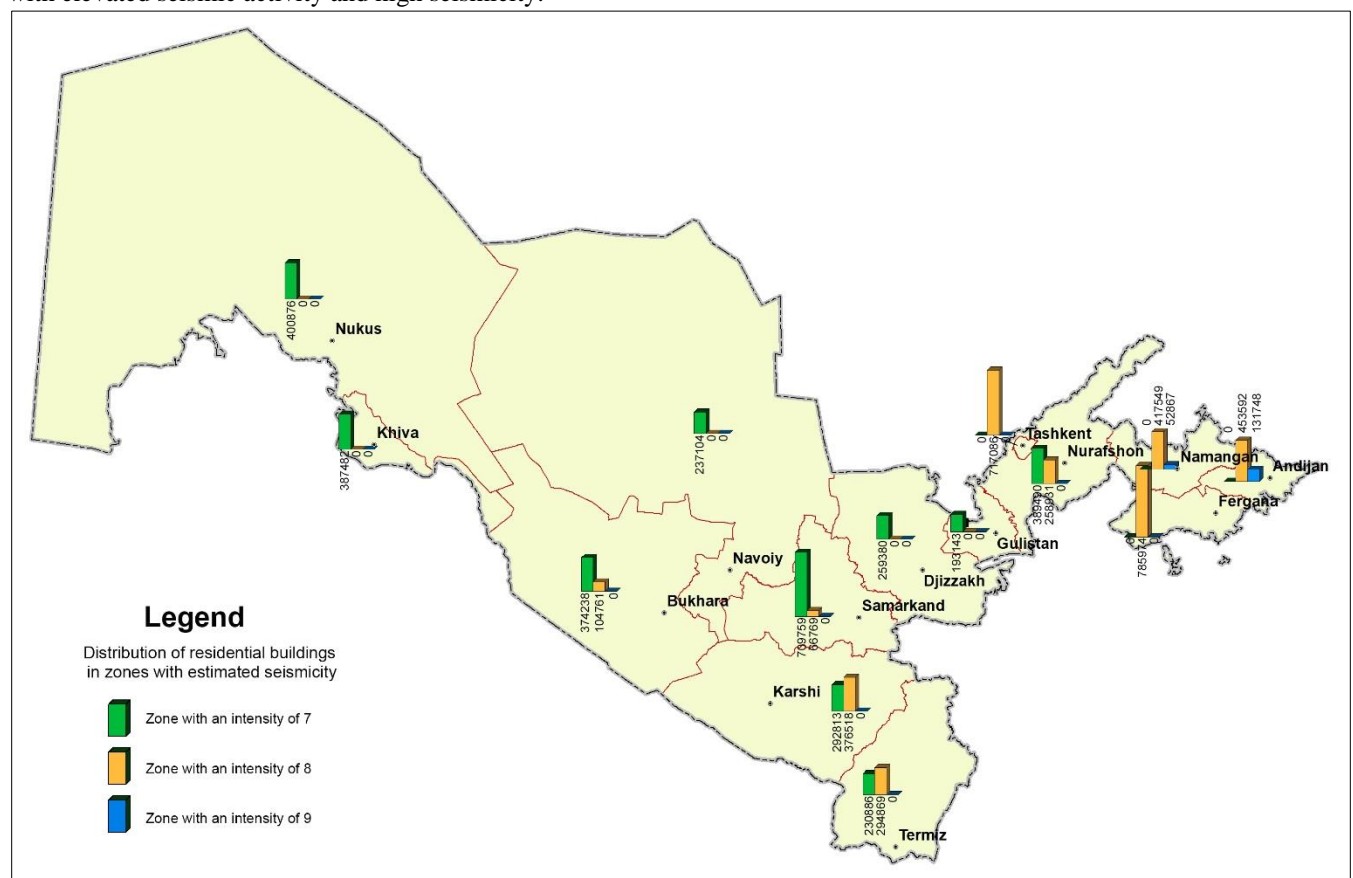

**Fig. 12:** Distribution of residential buildings based on the estimated seismicity of the territory in which the buildings are located within the administrative regions of the Republic of Uzbekistan

To develop a map of seismic risk of the territory of the Republic of Uzbekistan, several databases based on GIS platforms were created allowing systematization and evaluation of the regional distribution of information on seismic hazards, number of buildings and the material of the structural system, coefficient of the seismic vulnerability of buildings, cadastral value of buildings, etc.

The developed map of seismic risk of the territory of Republic of Uzbekistan is based on the assessment of probable economic losses within administrative regions, depending on the combination of seismic hazard factors, seismic vulnerability and concentration of values. It is important to emphasize that the level of seismic hazard used in the calculation of physical and economic damage corresponds to a 90% probability of not exceeding of seismic impacts for 50 years, which corresponds to an average return period of 475 years. This study is limited to the use of the return period of 475 years because this level of probability is generally accepted standard in seismic hazard assessment during the design and construction of conventional buildings and structures. Of course, considering a different probability, the level of danger and estimates of damage and potential losses may differ from the data presented.

The present study covered only the assessment of direct economic losses that may be caused by structural damage to residential buildings as a result of seismic events. At the same time, given that residential buildings predominate in the development of cities and administrative districts of the Republic of Uzbekistan, the presented results could serve as a clear reference for a comparative analysis of the seismic risk in various administrative districts.

Below is a small-scale map of the seismic risk in the territory of the Republic of Uzbekistan with an assessment of the probability of economic damage (Fig. 13) within the administrative districts at the maximum level of seismic impacts for the return period of T=475 years.

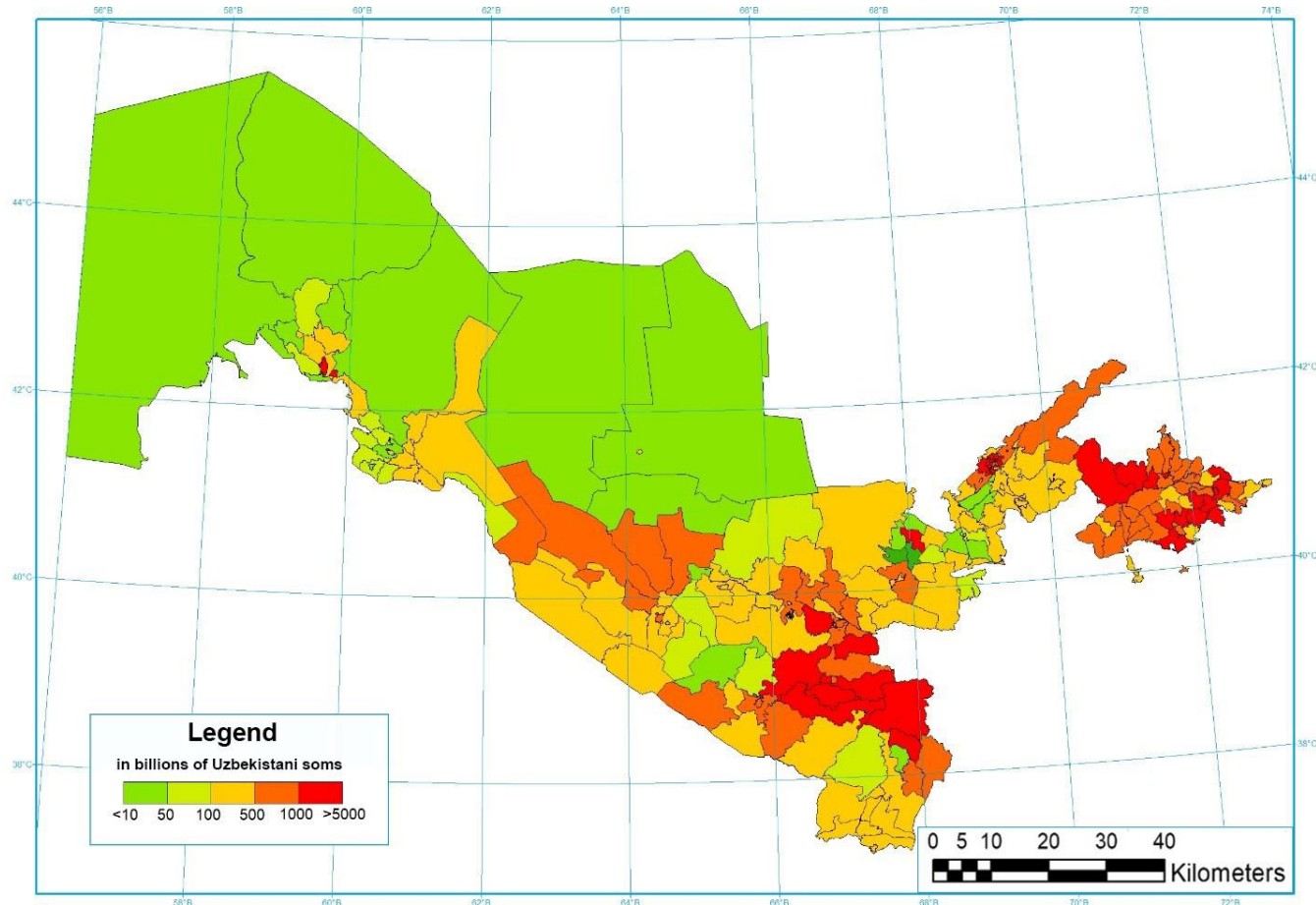


**Fig. 13:** Seismic risk assessment (in billions Uzbekistani soms) by administrative regions of the Republic of Uzbekistan.

## 4. Conclusions

Based on the study of geomorphological and geological structures and changes in the composition of 10-meter soil strata,
peculiarities of changes in engineering-geological conditions and seismic resistance of soils within the territory of Uzbekistan
have been identified. Additionally, for the first time in Uzbekistan, a seismic intensity increment map has been compiled with
a scale of 1:1000000.

Using the seismic zoning maps of the country (OSR-2017) for a 90% probability of not exceeding seismic effects over a 50-
year period and seismic intensity increments, a small-scale (1:1000000) schematic map of seismic intensity for the territory of
the republic has been developed. The seismicity of the territory has been refined based on the soil categories and their seismic
properties.

At the national level, as of February 1, 2021, a systematic electronic database has been created, containing information on
7135881 real estate properties, specifically residential buildings. Each property has been grouped based on its construction
type and coordinates in relation to administrative districts. This comprehensive database has been established to facilitate the
quantitative assessment of potential building damage during strong earthquakes, enabling the identification of preventive
measures to mitigate possible losses.

Based on the compiled schematic map of seismic intensity for the territory of Uzbekistan and the vulnerability functions
established for each construction type, the seismic vulnerability of the developed areas within the administrative districts has
been determined. The values of seismic vulnerability for the administrative districts fall within the following ranges: 0-0.15;
0.16-0.3; 0.31-0.45; 0.46-0.6; 0.61-0.75.

Comparison of the calculations and observational data for the damage caused by real past earthquakes reveals a suitable
agreement, indicating the correctness of the developed models and the efficiency of the calculation algorithms, which, in
combination with operational seismological information, could also be used to estimate losses due to earthquakes occurring in
real time.

Seismic vulnerability analysis and assessment were conducted using GESI_Program. Vulnerability models built depending on
the construction types of residential buildings characterized the vulnerability of residential buildings in all administrative
regions of Uzbekistan, which are subsequently considered as calculation cells. To assess the magnitude of potential damage in
monetary terms, cost indicators of the restoration of residential buildings were used. Seismic impacts were considered within
the framework of the project in the form of a probabilistic seismic hazard map. This approach made it possible to conduct a
comparative analysis of seismic risk distribution throughout the Republic of Uzbekistan.

When compiling a seismic risk map of the territory of the Republic of Uzbekistan, an administrative region was chosen as the territorial unit. This occurs because the scale of the study (1:1000000) does not allow for a detailed presentation of the existing database related to seismic hazard assessment, distribution of typical buildings, vulnerability assessment, etc.

The results obtained are presented in the form of maps showing the spatial distribution of possible damage to residential development and direct economic losses caused by this damage in all administrative regions of the Republic of Uzbekistan.

The territory of the Republic of Uzbekistan is characterized, on one hand, by a relatively high level of seismic hazards and on the other hand, by a relatively high concentration of residential buildings with low seismic resistance. Thus, possible future seismic events in the territory represent a typical high-probability problem with a potentially high level of losses. The obtained results and map of seismic risk could serve as a basis for the development of plans and measures to reduce the existing level of risk and prevent the catastrophic consequences of future earthquakes.

The present study covered only the estimation of direct economic losses of residential buildings in the Republic of Uzbekistan. At the same time, given that residential buildings predominate in the development of cities and towns in Uzbekistan, the presented results could serve as a clear reference for a comparative risk analysis throughout the Republic of Uzbekistan.

**Author contributions**

All of the authors contributed to the process of writing and verifying the research work and analyzed the results.

**Competing interests**

The authors declare that they have no known competing financial interests or personal relationships that could have appeared to influence the work reported in this paper.

**Disclaimer**

Publisher's note: Copernicus Publications remains neutral with regard to jurisdictional claims in published maps and institutional affiliations.

**Acknowledgements**

This work was funded by the Academy of Sciences of the Republic of Uzbekistan and Project "Regionally consistent risk assessment for earthquakes and floods and selective landslide scenario analysis for strengthening financial resilience and accelerating risk reduction in Central Asia" EU-funded Program "Strengthening Financial Resilience and Accelerating Risk Reduction in Central Asia" (SFRARR) was funded by the European Union and implemented by World Bank. We sincerely thank all the project team members, in particular Dr. Sergey Tyagunov and Dr. Paola Ceresa and the World Bank specialists, in particular Stuart Alexander Fraser and Madina Nizamitdin, for their constructive contribution to the project.

**Financial support**

This project was developed by the Institute of Seismology of the Academy of Sciences of the Republic of Uzbekistan at the request of the Academy of Sciences of the Republic of Uzbekistan

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
