# Peer review of "Regional seismic risk assessment based on ground conditions in Uzbekistan"

_Natural Hazards and Earth System Sciences, 2023_

## Community Comment (CC1)

Response to the Referee comments on the article "Regional seismic risk assessment based on ground conditions in Uzbekistan".

Thank you for careful consideration of the article.

1.      The article discusses the outcomes of developing GIS-platforms for seismic risk assessment in Uzbekistan. The significance of this publication is unquestionable. Nevertheless, in the reviewer's view, the authors have not effectively organized the information pertaining to the initial data used for risk assessments, nor have they adequately described the process for determining the final risk values. The text of the article is poorly structured, containing many introductory sentences, while there are no descriptions of specific stages of development of new maps. The article does not reveal the novelty of taking into account the ground conditions indicated in the title. The described changes in ground conditions accounting (135) are not used further and are not described. Furthermore, there are numerous inaccuracies within the article's text, tables and figures provided do not adhere to the standards expected in scientific publications.

Answer: We have corrected all reviewer's remarks and changed the structure of the article and eliminated the ambiguities.

2.      Table 1 is redundant. The text suggests that it includes events with magnitudes greater than or equal to 7, which does not align with the table's actual content. Additionally, there is no information regarding the type of magnitude used, and inconsistencies exist in the spelling of the same names. The date of the event 1924 is not provided.

Answer: In the table there are earthquakes with the same name, but these events took place in the same place at different times. We included dates in the table

3.      The title of the second section should be changed to "Data and methods"

Answer: Corrected

4.      101-102 - missing references.
5.      102 – The principle of division of the territory into 12 districts is not described. There is also no description of the division into sub-regions and sections.

Answer: we have removed 12 districts from the text. The map itself is divided into 14 districts by lithologic composition.

6.      Figure 1 should be modified. Only the demonstration areas and the legend should be shown. All information about the map should be given in the figure caption.

Answer: Done

7.      Figure 2 - see comments on Figure 1.

Answer: Corrected

8.      Figure 3 is not referenced in the text, and the panels within the figure remain undescribed. The panels essentially replicate maps found in other figures.

Answer: We corrected the numeration and inconsistencies

9.      Figure 4 - see comments on Figure 1. Figure 4 may be shown in conjunction with Figure 2. In this case it will be convenient for the reader to compare them

10.    The color code of intensity in Fig. 2 and Fig. 4 must be the same.

Answer: Corrected

11.    Changes in the definition of intensity should be described in more detail. For example, by presenting a table of area for one and the other seismic hazard maps.

Answer: Ratios between seismic hazard map and seismic hazard map considering ground conditions in percentage.

12.    GESI_Program - missing references

Answer: Corrected

13.    "Damage characteristics of buildings" - table it.

Answer: We have included table.

14.    240 - The vulnerability functions used should be cited. If they are presented in Fig. 5, this should be indicated. The article does not specify (except for Fig. 5) the ratio of peak acceleration and macroseismic intensity used. A correspondence table or conversion formula (with references) is needed

Answer: We have included the citation on vulnerability function. We have included a conversion equation with reference.

15.    252 "GESI_Program and experimental data of Sh. Khakimov" - missing references

Answer. We have corrected the references

16.    305, Figure 9 - PGA needs to be in $m/s^2$ as on Figure 5. The grading of the PGA in Fig. 8 is not clear. It would seem that it should coincide with the one in Fig. 5 and, accordingly, with the intervals corresponding to the seismic intensity values.; EMS-98 - missing references

Answer. We have corrected the figure and included the reference.

17.     Since administrative divisions are difficult to present to the general reader, the information in Figure 11 should either be presented in the form of a map or population numbers should be given instead of/along with the names of administrative divisions. see comments on Figure 9

Answer: It is not possible to separate population instead of/along with the names of administrative units because there are different PGA for each city.

[Figure]

**Figure 11:** Distribution of residential buildings in the territories with different seismic effects within the administrative regions in Uzbekistan.

18.     335-340 - Technical information is redundant. If the database is open, a link to it should be provided. If it is closed for public access, this should also be indicated.

Answer: We have removed the redundant technical information and Figures 13 and 14.

19.     360-390 The section provides a map of seismic risk. It is not clear what the authors meant by "Probable seismic damage" in the title of the paragraph. Since the title of the article contains new seismic hazard estimates, seismic risk estimates based on the previous seismic hazard map should be given for comparison.

Answer: The seismic risk map is calculated only considering the ground conditions

---

## Author Comment (AC1)

Response to comments of the reviewer

Thank you for the thorough review of our article. We revised the paper and increased the quality of the paper.

Please note that some revisions were shown with the function of tracking changes while other changes were highlighted with yellow marker.

1. Line 25-26. As of January 1, 2022, the permanent population of Uzbekistan reached 35 271 276 people. Currently, approximately half of all Uzbekistan citizens (17.9 million people) live in urban areas and 17.4 million people live in rural areas (Please add reference).

*We have added the reference to the reference list.*

*(https://countrymeters.info/ru/Uzbekistan#population_densit).*

2. Line 27. … earthquakes with a magnitude of M ≥ 7 (Please indicate the type of magnitude)

*We have used the Local Magnitude Scale $M_L$.*

3. In the Lines 27-29 the sentence : At the territory of Uzbekistan and adjacent regions, both during the historical period and recent years, earthquakes with a magnitude of M ≥ 7 and an intensity at the epicentre I0 reaching 9–10 according to the MSK-64 scale have been recorded (Table 1). It is confusing since the data constitute Table 1 are with also smaller magnitudes than 7 even 5… Please explain. I guess this is not even the full earthquake catalog of Uzbekistan but some extract…

*Yes, it does not encompass the entire earthquake catalog. We have utilized the earthquake catalog available at the Institute of Seismology in Uzbekistan. Our objective was to carefully curate a selection of the most impactful earthquakes from various locations over the past century. This effort aims to illustrate that Uzbekistan is indeed prone to seismic activity.*

4. Line 28. … have been recorded (Table 1) (Are all of those earthquakes stated in Table 1 are really instrumentally recorded? Even the historical ones? Please clarify.)

*We have added asterisk\* to distinguish historical earthquakes.*

3. Line 29-35. The geological structure of Uzbekistan is very diverse, but the territory basically consists of two tectonic structures of the Tien Shan orogenic region and Turan plate. The current state of relief in the territory of Uzbekistan was preceded by long difficult stages. In the territory of Uzbekistan, tectonic movements are actively continuing nearly everywhere. In the geological history of Uzbekistan, roughout all stages of development, in particular, in the formation of the modern structural plan, faults, especially zones of deep faults, played an important role. These faults transect the entire Earth's crust, often penetrate into the mantle and are the natural boundaries of large structural elements. (Please add reference/s in relation to statements about contemporary geology and tectonic of territory of Uzbekistan).

*We have added the reference in relation to statements about contemporary geology and tectonic of the territory of Uzbekistan (V.I. Ulomov et al., 1990).*

5. Line 32. The current state of relief in the territory of Uzbekistan was preceded by long difficult stages. (What does that mean? Please revise the sentence and explain better).

*The sentence was there due to improper translation into English. We have removed it.*

6. Line 35. These faults influence disaster preparedness and risk reduction activities. (Please rewrite as faults cannot influence any activities. Maybe seismic conditions is better term…).

*Yes, you are right. Thank you for pointing it out. We have corrected it.*

7. Line 35-39. One of the challenges in assessing seismic risk is considering the determination of soil conditions in the modification of seismic effects on the Earth's surface. Therefore, one of the tasks of this study is to investigate the geological environment and the patterns of seismic wave propagation through it. This is because this effect is directly dependent on the structure and depth of the geological and lithological differences of the rock formations comprising it. (Please revise, improve English)

*Thank you for your comment. We have revised the paragraph and improved English.*

8. Line 40. Table 1. Date format to be extracted year, day, month in separate columns. Name? What that means? Name of the closest site to the epicenter or maybe region? Please explain and include the explanation in the text. For M please indicate type of magnitude (ML, Mw… or other).

*We have corrected the name of table columns accordingly.*

9. Line 44-61. Repetitions of sentences noted. Mistakes in some references (ex. Trendafiloski and Milutin (2004)… should be Milutinovic…). Missing new and state of the art refences and worldwide initiatives in the domain of seismic risk (ex. GEM initiative or similar). In this part it is necessary to include references focused on the Central Asia Region and on national level with proper comments from the authors.

*We have removed repetitions, corrected the references and added recent literature.*

10. Line 66. Is the risk assessment comprising only residential building portfolio? Please clarify and explain.

*For our study we use the database of the cadastral agency of Uzbekistan, and it has only residential buildings and their cost estimation, which is necessary for seismic risk assessment*

11. Line 70-73 The developed seismic risk analysis algorithm used the capabilities of GIS, combining data on the spatial distribution of seismic hazards, vulnerability of buildings, geographical location of residential buildings, and values, i.e., cadastral value of buildings at risk of damage and loss, in a layer-by-layer manner. (Need revision and better explanation).

*We revised that part and removed unclear explanations.*

12. Line 73. GESI_Program (https://iisee.kenken.go.jp/net/saito/gesi_program/index.html). What was the idea of using this „quite old" nearly 25 years old tool despite existence of

other state-of-the-art tools and softwares for seismic risk calculations (ex. Open Quake, HAZUS, Selena, CAPRA, ELER, and others…)? Please explain.

This is a pilot project for Uzbekistan, and we wanted to avoid complications and keep the methodology simple and straightforward. Moreover, we had experience using GESI before. Therefore, we used this comparatively old tool for this study. In further studies and projects, we are going to use more modern tools, including software that was listed by you.

13. Line 87-93. Please include the web site links to the mentioned institutions in the footnote.

*We have added the web site links to the mentioned institutions in the footnote.*

14. Line 101-101. Please refer the mentioned works of G.A. Mavlyanov, A.I. Islamov, P.M. Karpov, S.M. Kasymov, R.F. Kirsanova, A.M. Khudaibergenov, M.Sh. Shermatov, K.P. Pulatov in correct manner and include them in reference list

*We have revised the mentioned works in a correct manner and included them in the reference list*

15. Line 105 Figure 1. Please add reference related to this figure/map

[Figure]

*We have added the reference to the figure/map.*

16. Line 132. Please explain term "average soil" and relate it so soil category.

*We removed the confusing term and explained our methodology with other words.*

Line 107-120. Please support with the references.

*We have added the reference.*

17. Chapter 2.2. Is very confusing. Must be rewritten as a whole, better explained and accordingly referenced.

*We have revised chapter 2.2, added references and tried to provide a better explanation of our work.*

18. Line 173 - 175. Vulnerability functions for the identified structural building types within the territory of the Republic of Uzbekistan were developed using the "GESI_Program", which is a computer program based on the assessment of structural damage under 175 specified seismic events (see Fig. 5). (Please explain how they are developed?)

*To establish vulnerability functions in the 'GESI_Program,' several parameters are considered: type of construction material, design quality, construction quality, seismic strength. Following this, the 'GESI_Program' will calculate the vulnerability functions.*

19. Line 182. The vulnerability index for the city of Tashkent in the experiment did not exceed 10% of the total (Probably this is result of RADIUS project. Please add reference for this statement).

*We have added the reference for this statement.*

20. Line 191-193. As of February 1, 2021, at the republican level, 7,135,881 residential buildings were analyzed and systematized with a total area of 4.4 billion square meters. These buildings were categorized by their structural types and aggregated by administrative regions (By whom? By this research or?)

*It was done by the employees of the Institute of Seismology (Uzbekistan). We have added this information to the text.*

21. Line 195, Table 2. It is stated that the buildings are classified according to structural types, which is not true but according to material of structural system (Please explain and clarify).

*We meant the material of structural system. We have revised this part.*

22. Chapter 3.1. Title should be revised and reflect the content.

*We have revised the title.*

23. Line 286. … individual houses (80.1%) and multi-story residential buildings (19.9%). What means individual houses… only ground floor or? Accordingly, that means multy-story (G+1 up to … ) Please clarify.

*Individual houses are 1 or 2 story buildings, and residential buildings are buildings, which have many apartments within the building. We have clarified that part in the text.*

24. Chapter 3. Data and statistics presented is better also to be shown in spatial (GIS) manner.

*We have added a new figure and showed the spatial distribution.*

25. Chapter 3.2. The whole section needs to be seriously rewritten and better explained.

*Since there is a similar information in the next chapter, we removed that chapter.*

26. Chapter 3.3. Title… Probable…? You mean probabilistic? Probabilistic seismic damage and risk assessment?

*Yes, we meant probabilistic seismic damage and risk assessment. We revised that part.*

27. Chapter 3.3. The content should also be seriously rewritten. It is a summary of previous sections. Why only one return period is considered in the study (475 y)?

*We chose a return period of T=475 years, as it is commonly used in similar studies in Germany, Italy, and other countries to assess seismic risk. Also, we have revised the chapter.*

---

## Author Comment (AC2)

**Response to the Referee comments on the article "Regional seismic risk assessment based on ground conditions in Uzbekistan".**

Many thanks to the two reviewers for reviewing our article in detail and expressing their feedback.

Authors answered fully all the comments and questions posed. The manuscript has been modified accordingly. We have corrected all your comments. The following is the corrected comments.

**Reviewer #1:**

1.      The article discusses the outcomes of developing GIS-platforms for seismic risk assessment in Uzbekistan. The significance of this publication is unquestionable. Nevertheless, in the reviewer's view, the authors have not effectively organized the information pertaining to the initial data used for risk assessments, nor have they adequately described the process for determining the final risk values. The text of the article is poorly structured, containing many introductory sentences, while there are no descriptions of specific stages of development of new maps. The article does not reveal the novelty of taking into account the ground conditions indicated in the title. The described changes in ground conditions accounting (135) are not used further and are not described. Furthermore, there are numerous inaccuracies within the article's text, tables and figures provided do not adhere to the standards expected in scientific publications.

Answer: We have corrected all reviewer's remarks and changed the structure of the article and eliminated the ambiguities.

2.      Table 1 is redundant. The text suggests that it includes events with magnitudes greater than or equal to 7, which does not align with the table's actual content. Additionally, there is no information regarding the type of magnitude used, and inconsistencies exist in the spelling of the same names. The date of the event 1924 is not provided.

Answer: In the table there are earthquakes with the same name, but these events took place in the same place at different times. We included dates in the table

3.      The title of the second section should be changed to "Data and methods"

Answer: Corrected

4.      101-102 - missing references.
5.      102 – The principle of division of the territory into 12 districts is not described. There is also no description of the division into sub-regions and sections.

Answer: we have removed 12 districts from the text. The map itself is divided into 14 districts by lithologic composition.

6.      Figure 1 should be modified. Only the demonstration areas and the legend should be shown. All information about the map should be given in the figure caption.

Answer: Done

7.      Figure 2 - see comments on Figure 1.

Answer: Corrected

8.      Figure 3 is not referenced in the text, and the panels within the figure remain undescribed. The panels essentially replicate maps found in other figures.

Answer: We corrected the numeration and inconsistencies

9.      Figure 4 - see comments on Figure 1. Figure 4 may be shown in conjunction with Figure 2. In this case it will be convenient for the reader to compare them

10.     The color code of intensity in Fig. 2 and Fig. 4 must be the same.

Answer: Corrected

11.     Changes in the definition of intensity should be described in more detail. For example, by presenting a table of area for one and the other seismic hazard maps.

Answer: Ratios between seismic hazard map and seismic hazard map considering ground conditions in percentage.

12.     GESI_Program - missing references

Answer: Corrected

13.     "Damage characteristics of buildings" - table it.

Answer: We have included table.

14.     240 - The vulnerability functions used should be cited. If they are presented in Fig. 5, this should be indicated. The article does not specify (except for Fig. 5) the ratio of peak acceleration and macroseismic intensity used. A correspondence table or conversion formula (with references) is needed

Answer: We have included the citation on vulnerability function. We have included a conversion equation with reference.

15.     252 "GESI_Program and experimental data of Sh. Khakimov" - missing references

Answer. We have corrected the references

16.     305, Figure 9 - PGA needs to be in $m/s^2$ as on Figure 5. The grading of the PGA in Fig. 8 is not clear. It would seem that it should coincide with the one in Fig. 5 and, accordingly, with the intervals corresponding to the seismic intensity values.; EMS-98 - missing references

Answer. We have corrected the figure and included the reference.

17.      Since administrative divisions are difficult to present to the general reader, the information in Figure 11 should either be presented in the form of a map or population numbers should be given instead of/along with the names of administrative divisions. see comments on Figure 9

Answer: Figure 11 shows the distribution of residential buildings in areas with different seismic impacts within the administrative districts of Uzbekistan. We have redrawn it to show the distribution of residential buildings by regions of Uzbekistan.

[Figure]

**Fig. 11:** Distribution of residential buildings by regions in Uzbekistan

18. 335-340 - Technical information is redundant. If the database is open, a link to it should be provided. If it is closed for public access, this should also be indicated.

Answer: We have removed unnecessary technical information, redrawn, and included Figures 12 and 13 in the text.

*To assess the seismic risk within the context of the administrative districts of the Republic of Uzbekistan, it is necessary to take into account the share of the housing stock across all administrative districts, considering zones with different intensities. Figure 10 shows the share of residential buildings in Uzbekistan with different parameters of seismic vibrations (based on the OSR-2017 map with a probability of 90%). Considering spatial distribution of residential buildings by zones with different seismicity (based on the OSR-2017 map with a probability of 90%), it can be seen that southern and eastern parts of the country having an estimated seismicity of 8 and the most eastern part, which is Ferghana valley, has seismicity of 9 (Fig. 11). Cadastral value of residential buildings by administrative areas is also an important information for developing maps of seismic risk, as well as for the government that implementing policies for increasing the seismic resilience of buildings and structures. Figure 12 shows the cadastral value of housing stock within the Republic of Uzbekistan and its administrative areas.*

19. 360-390 The section provides a map of seismic risk. It is not clear what the authors meant by "Probable seismic damage" in the title of the paragraph. Since the title of the article contains new seismic hazard estimates, seismic risk estimates based on the previous seismic hazard map should be given for comparison.

Answer: The seismic risk map is calculated only considering the ground conditions

**Reviewer #2:**

1.     Line 25-26. As of January 1, 2022, the permanent population of Uzbekistan reached 35 271 276 people. Currently, approximately half of all Uzbekistan citizens (17.9 million people) live in urban areas and 17.4 million people live in rural areas (Please add reference).

*We have added the reference to the reference list.*

*(https://countrymeters.info/ru/Uzbekistan#population_densit).*

2.     Line 27. … earthquakes with a magnitude of M ≥ 7 (Please indicate the type of magnitude)

*We have used the Local Magnitude Scale $M_L$.*

3.     In the Lines 27-29 the sentence : At the territory of Uzbekistan and adjacent regions, both during the historical period and recent years, earthquakes with a magnitude of M ≥ 7 and an intensity at the epicentre I0 reaching 9–10 according to the MSK-64 scale have been recorded (Table 1). It is confusing since the data constitute Table 1 are with also smaller magnitudes than 7 even 5… Please explain. I guess this is not even the full earthquake catalog of Uzbekistan but some extract…

*Yes, it does not encompass the entire earthquake catalog. We have utilized the earthquake catalog available at the Institute of Seismology in Uzbekistan. Our objective was to carefully curate a selection of the most impactful earthquakes from various locations over the past century. This effort aims to illustrate that Uzbekistan is indeed prone to seismic activity.*

4.     Line 28. … have been recorded (Table 1) (Are all of those earthquakes stated in Table 1 are really instrumentally recorded? Even the historical ones? Please clarify.)

*We have added asterisk\* to distinguish historical earthquakes.*

3.     Line 29-35. The geological structure of Uzbekistan is very diverse, but the territory basically consists of two tectonic structures of the Tien Shan orogenic region and Turan plate. The current state of relief in the territory of Uzbekistan was preceded by long difficult stages. In the territory of Uzbekistan, tectonic movements are actively continuing nearly everywhere. In the geological history of Uzbekistan, roughout all stages of development, in particular, in the formation of the modern structural plan, faults, especially zones of deep faults, played an important role. These faults transect the entire Earth's crust, often penetrate into the mantle and are the natural boundaries of large structural elements. (Please add reference/s in relation to statements about contemporary geology and tectonic of territory of Uzbekistan).

*We have added the reference in relation to statements about contemporary geology and tectonic of the territory of Uzbekistan (V.I. Ulomov et al., 1990).*

5.     Line 32. The current state of relief in the territory of Uzbekistan was preceded by long difficult stages. (What does that mean? Please revise the sentence and explain better).

*The sentence was there due to improper translation into English. We have removed it.*

6.     Line 35. These faults influence disaster preparedness and risk reduction activities. (Please rewrite as faults cannot influence any activities. Maybe seismic conditions is better term…).

*Yes, you are right. Thank you for pointing it out. We have corrected it.*

7.      Line 35-39. One of the challenges in assessing seismic risk is considering the determination of soil conditions in the modification of seismic effects on the Earth's surface. Therefore, one of the tasks of this study is to investigate the geological environment and the patterns of seismic wave propagation through it. This is because this effect is directly dependent on the structure and depth of the geological and lithological differences of the rock formations comprising it. (Please revise, improve English)

*Thank you for your comment. We have revised the paragraph and improved English.*

8.      Line 40. Table 1. Date format to be extracted year, day, month in separate columns. Name? What that means? Name of the closest site to the epicenter or maybe region? Please explain and include the explanation in the text. For M please indicate type of magnitude (ML, Mw… or other).

*We have corrected the name of table columns accordingly.*

9.      Line 44-61. Repetitions of sentences noted. Mistakes in some references (ex. Trendafiloski and Milutin (2004)… should be Milutinovic…). Missing new and state of the art refences and worldwide initiatives in the domain of seismic risk (ex. GEM initiative or similar). In this part it is necessary to include references focused on the Central Asia Region and on national level with proper comments from the authors.

*We have removed repetitions, corrected the references and added recent literature.*

10.     Line 66. Is the risk assessment comprising only residential building portfolio? Please clarify and explain.

*For our study we use the database of the cadastral agency of Uzbekistan, and it has only residential buildings and their cost estimation, which is necessary for seismic risk assessment*

11.     Line 70-73 The developed seismic risk analysis algorithm used the capabilities of GIS, combining data on the spatial distribution of seismic hazards, vulnerability of buildings, geographical location of residential buildings, and values, i.e., cadastral value of buildings at risk of damage and loss, in a layer-by-layer manner. (Need revision and better explanation).

*We revised that part and removed unclear explanations.*

12.     Line 73. GESI_Program (https://iisee.kenken.go.jp/net/saito/gesi_program/index.html). What was the idea of using this „quite old" nearly 25 years old tool despite existence of other state-of-the-art tools and softwares for seismic risk calculations (ex. Open Quake, HAZUS, Selena, CAPRA, ELER, and others…)? Please explain.

This is a pilot project for Uzbekistan, and we wanted to avoid complications and keep the methodology simple and straightforward. Moreover, we had experience using GESI before. Therefore, we used this comparatively old tool for this study. In further studies and projects, we are going to use more modern tools, including software that was listed by you.

13.     Line 87-93. Please include the web site links to the mentioned institutions in the footnote.

*We have added the web site links to the mentioned institutions in the footnote.*

14. Line 101-101. Please refer the mentioned works of G.A. Mavlyanov, A.I. Islamov, P.M. Karpov, S.M. Kasymov, R.F. Kirsanova, A.M. Khudaibergenov, M.Sh. Shermatov, K.P. Pulatov in correct manner and include them in reference list

*We have revised the mentioned works in a correct manner and included them in the reference list*

15. Line 105 Figure 1. Please add reference related to this figure/map

[Figure]

*We have added the reference to the figure/map.*

16. Line 132. Please explain term "average soil" and relate it so soil category.

*We removed the confusing term and explained our methodology with other words.*

Line 107-120. Please support with the references.

*We have added the reference.*

17. Chapter 2.2. Is very confusing. Must be rewritten as a whole, better explained and accordingly referenced.

*We have revised chapter 2.2, added references and tried to provide a better explanation of our work.*

18. Line 173 - 175. Vulnerability functions for the identified structural building types within the territory of the Republic of Uzbekistan were developed using the "GESI_Program", which is a computer program based on the assessment of structural damage under 175 specified seismic events (see Fig. 5). (Please explain how they are developed?)

*To establish vulnerability functions in the 'GESI_Program,' several parameters are considered: type of construction material, design quality, construction quality, seismic strength. Following this, the 'GESI_Program' will calculate the vulnerability functions.*

19.     Line 182. The vulnerability index for the city of Tashkent in the experiment did not exceed 10% of the total (Probably this is result of RADIUS project. Please add reference for this statement).

*We have added the reference for this statement.*

20.     Line 191-193. As of February 1, 2021, at the republican level, 7,135,881 residential buildings were analyzed and systematized with a total area of 4.4 billion square meters. These buildings were categorized by their structural types and aggregated by administrative regions (By whom? By this research or?)

*It was done by the employees of the Institute of Seismology (Uzbekistan). We have added this information to the text.*

21.     Line 195, Table 2. It is stated that the buildings are classified according to structural types, which is not true but according to material of structural system (Please explain and clarify).

*We meant the material of structural system. We have revised this part.*

22.     Chapter 3.1. Title should be revised and reflect the content.

*We have revised the title.*

23.     Line 286. … individual houses (80.1%) and multi-story residential buildings (19.9%). What means individual houses… only ground floor or? Accordingly, that means multy-story (G+1 up to … ) Please clarify.

*Individual houses are 1 or 2 story buildings, and residential buildings are buildings, which have many apartments within the building. We have clarified that part in the text.*

24.     Chapter 3. Data and statistics presented is better also to be shown in spatial (GIS) manner.

*We have added a new figure and showed the spatial distribution.*

[Figure]

**Fig. 11:** Distribution of residential buildings by regions in Uzbekistan

25.     Chapter 3.2. The whole section needs to be seriously rewritten and better explained.

*Since there is a similar information in the next chapter, we removed that chapter.*

*26.     Chapter 3.3. Title… Probable…? You mean probabilistic? Probabilistic seismic damage and risk assessment?*
*Yes, we meant probabilistic seismic damage and risk assessment. We revised that part.*

*27.     Chapter 3.3. The content should also be seriously rewritten. It is a summary of previous sections. Why only one return period is considered in the study (475 y)?*
*We chose a return period of T=475 years, as it is commonly used in similar studies in Germany, Italy, and other countries to assess seismic risk. Also, we have revised the chapter.*

---

## Author Response (AR1)

Thank you for kindly reviewing our paper and checking our responses to the reviewers. There were some issues with the structure of the paper and some unclear parts, and it was good that you pointed them out. We think that with your help the quality of the paper has increased.

**Comment:** Please explain carefully the methodology that you use and make sure you disambiguate what you mean by 'probable seismic damage'.

**Answer:** We have included the more detailed explanation of the methodology in the text.

By the term "probable seismic damage" we meant the estimated economic damage that would occur in a specific area in the event of an earthquake. We agree that it was unclear; therefore, we removed this phrase and kept a simpler explanation. Also, for clarity, we have corrected the title of Chapter 3.2 from "Probable Seismic Risk Assessment" to "Seismic Risk Assessment".

**Comment:** Please discuss how the buildings classification that you used compares with other building typologies defined for Uzbekistan and at the regional scale for Central Asia (e.g. EMCA).

**Answer:** We have included a table (Table 6) which compares the local classification (which we used in our study) with the EMCA classification.

**Table 6.** Classification of buildings in Tashkent according to the vulnerability index

| | Our classification | EMCA | | |
|---|---|---|---|---|
| | | EMCA Classification | Subtype | Description |
| 1 | Adobe (local) | EMCA4 | ADO | Adobe structures |
| 2 | Masonry | EMCA1 | CM | Brick masonry of a complex structure |
| 3 | Wooden | EMCA5 | WOOD1 | Wooden structure, load-bearing frames with connections |
| | | | WOOD2 | Wooden structure, wooden frame, and adobe infill |
| 4 | Concrete | EMCA2 | All subtypes EMCA2 | All descriptions of EMCA2 subtypes |
| | | EMCA3 | All subtypes EMCA3 | All descriptions of EMCA3 subtypes |
| 5 | Metal frame | EMCA6 | STEEL | Steel structures |

**Comment:** Carefully rewrite section 3. Please make sure that session 3 only contains results and discussion, while data sources and methods should be included in section 2. For example, the number and type of buildings are necessary to perform the risk assessment so they should be introduced before you estimate the risk. Also, 3.1 speaks about asset values but you provide figures in terms of number of buildings, not economic value, which is mentioned in section 3.2.

**Answer:** Thank you for your comment. We have revised section 3 and put the data sources and methods in section 2. Also, we have added the figure in terms of economic value which is mentioned in section 3.2 (Fig. 13).

**Comment:** Please make sure the figure captions contain enough explanation for the reader to understand them and consider merging figures.

**Answer**: We have revised the figure captions to have a better explanation. But we think that merging figures will negatively affect the flow of discussion.

**Comment:** I think your response to the last comment of reviewer #1 is unclear ("The seismic risk map is calculated only considering the ground conditions"). Please specify in the manuscript what you mean and discuss how your results compare to past risk assessments done in the region.

**Answer:** We have specified in the manuscript the idea of compiling the seismic risk map based on the ground conditions and have added the past risk assessments and similar studies done in the region.

---

## Author Response (AR2)

Thank you for thoroughly checking the paper and given comments. After addressing your comments, the quality of paper has been increased.

1) *In particular, I think the introduction could cover a bit more the state of the art (methods, tools and approaches that exist in literature, not only for the study area but in general).*

**Answer:** We have included several references in the introduction section.

As of today, Peresan et al. (2023), Poggi et al. (2021), Bragato et al. (2020), Petrovic et al. (2022, 2023), Scaini et al. (2021, 2023), Bhochhibhoya et al. (2022), and Xin et al. (2021) explore contemporary methods for assessing seismic risk and hazard using modern information technologies. Bhochhibhoya et al. (2022) integrated earthquake risk assessments with vulnerability parameters (social and economic factors) in Nepal. The calculation of the Social Vulnerability Index (SoVI) used a principal component analysis method. OpenQuake, based on classical Probabilistic Seismic Hazard Analysis (PSHA), was utilized for calculating annual average losses from engineering risk. In the work of Peresan et al. (2023), the focus is on data collection about buildings through crowdsourcing and distance learning for new opportunities to engage students in seismic risk reduction.

2) *Also, following the reviewers' suggestions, some figures (e.g. 11 and 12) and tables (e.g. table 1) are somehow redundant and some tables (e.g. table 5 and 6) could be merged together, improving the paper readability.*

**Answer:** We have removed redundant figures and tables, and merged tables 5 and 6, which improved the paper readability.

Table 4. Residenttial building taxonomy

| | Our classification | Classification of buildings in Uzbekistan | EMCA Classification |
|---|---|---|---|
| 1 | Adobe (local) | Residential buildings constructed from local low-strength materials (without anti-seismic measures) | EMCA4 |
| | | One-story clay walls of the guvalyak and pakhsa types | |
| 2 | Masonry | Three- to five-storey frameless brick buildings with wooden floors constructed until 1958 | EMCA1 |
| | | One- to two-storey frameless brick walls with wooden floors | |
| | | Walls made of bricks, small concrete or natural stones; ceilings - prefabricated reinforced concrete | |
| | | Buildings with external load-bearing brick walls; internal - reinforced concrete frame elements | |
| | | Walls made of large blocks (concrete, vibro-brick, or reinforced vibro-brick panels) | |
| | | Reinforced concrete frame with brick filling | |
| 3 | Wooden | One- to two-storey wooden houses (chopped or panel) | EMCA5 |
| | | One- to two-storey wooden frames filled with raw bricks (sinch) | |
| 4 | Concrete | Prefabricated reinforced concrete frame made of linear elements with a welded joint in the zone of maximum effort, or the same with stiffening diaphragms in one direction (framework III of the IIS-04 series and their modifications) | EMCA2 |
| | | Large-panel walls without anti-seismic measures | |
| | | Walls of complex construction (with reinforced concrete inclusions); ceilings - prefabricated reinforced concrete | |

| | | Large panel walls | |
|---|---|---|---|
| | | Monolithic reinforced concrete frame | EMCA3 |
| | | Prefabricated reinforced concrete frame-braced frame with monolithic nodes, with stiffening diaphragms in two directions or stiffening cores | |
| | | Frame made of spatial elements (volumetric cross) with monolithic knots | |
| | | Frame made of spatial elements (volumetric cross) with monolithic knots | |
| | | Volumetric blocks per room | |
| 5 | Metal frame | Metal frame or frame with diaphragms (bonds) | EMCA6 |

*3) Finally, the conclusions could be polished a bit more so that the reader can understand which are the main findings and how they contribute to the current state of knowledge.*

**Answer:** We have polished the conclusion section and revised main findings and contribution to the current state of knowledge.

**4. Conclusions**

Based on the study of geomorphological and geological structure, as well as changes in the composition of the upper 10-meter soil strata, features of changes in engineering-geological conditions and seismic resistance of soils in the territory of Uzbekistan have been identified. Using seismic zoning maps of the country (OSR-2017) with a 90% probability of not exceeding seismic impacts over a 50-year period and considering seismic intensity increments, a microscale seismic intensity map (1:1 000 000) for the entire republic has been developed. The seismicity of the territory has been calculated, taking into account soil categories by their seismic properties. Seismically hazardous areas consist of different soil conditions, whereas the General Seismic Zoning (OSR) map considers average soil conditions. By meticulous consideration of soil conditions of the regions, the reliability of the assessment of seismic hazard in regions has been increased.

At the national level, as of February 1, 2021, a systematic electronic database has been created, containing information on 7135881 real estate properties, specifically residential buildings. Each property has been grouped based on its construction type and coordinates in relation to administrative districts. This comprehensive database has been established to facilitate the quantitative assessment of potential building damage during strong earthquakes, enabling the identification of preventive measures to mitigate possible losses.

Based on the compiled schematic map of seismic intensity for the territory of Uzbekistan and the vulnerability functions established for each construction type, the seismic vulnerability of the developed areas within the administrative districts has been determined. The values of seismic vulnerability for the administrative districts fall within the following ranges: 0-0.15; 0.16-0.3; 0.31-0.45; 0.46-0.6; 0.61-0.75. From these vulnerability values, it is possible to determine the degree of vulnerability for each region.

Seismic vulnerability analysis and assessment were conducted using GESI_Program. Vulnerability models built depending on the construction types of residential buildings characterized the vulnerability of residential buildings in all administrative regions of Uzbekistan, which are subsequently considered as calculation cells. To assess the magnitude of potential damage in monetary terms, cost indicators of the restoration of residential buildings were used. Seismic impacts were considered within the framework of the project in the form of a probabilistic seismic hazard map. This approach allows for a comparative analysis of the distribution of seismic risk across seismically hazardous areas.

The present study covered only the estimation of direct economic losses of residential buildings in the Republic of Uzbekistan. At the same time, considering that residential construction predominates in the development of many states, the presented results can serve as a clear guide

for a comparative analysis of risks across the entire seismically hazardous territory. The obtained results and such seismic risk maps can serve as a basis for the development of plans and measures to reduce the existing level of risk and prevent catastrophic consequences of future earthquakes for government agencies dealing with emergency situations.

4) *Please explain carefully the methodology that you use and make sure you disambiguate what you mean by 'probable seismic damage'.*

**Answer:** The seismic vulnerability of buildings was assessed based on the GESI_Program, which was developed during the RADIUS program (1999-2001). Vulnerability functions were developed according to the material of structural system. Based on this, map with spatial distribution of buildings having different vulnerabilities within the administrative regions of the Republic of Uzbekistan was compiled. We determined the percentage of building damage based on the vulnerability function, obtained the building inventory value from the Cadastral Agency of the Republic of Uzbekistan, and established the economic loss from the amount of damage that the building suffered from the scenario earthquake.

Probable seismic damage refers to the likelihood of socio-economic harm resulting from potential earthquakes, based on the calculated seismic hazard of the area and the vulnerability of buildings. Maps of seismic risk can be compiled by combining assessments of seismic hazard and vulnerability of buildings and structures in populated areas. These maps serve as the basis for estimating the expected damage in monetary terms, considering the cost of objects within the affected regions.

5) *Please discuss how the buildings classification that you used compares with other building typologies defined for Uzbekistan and at the regional scale for Central Asia (e.g. EMCA).*

The correlation of the used building classification with other building typologies (e.g. EMCA) is summarized in the table below.

| | Our classification | Classification of buildings in Uzbekistan | EMCA Classification |
|---|---|---|---|
| 1 | Adobe (local) | Residential buildings constructed from local low-strength materials (without anti-seismic measures) | EMCA4 |
| | | One-story clay walls of the guvalyak and pakhsa types | |
| 2 | Masonry | Three- to five-storey frameless brick buildings with wooden floors constructed until 1958 | EMCA1 |
| | | One- to two-storey frameless brick walls with wooden floors | |
| | | Walls made of bricks, small concrete or natural stones; ceilings - prefabricated reinforced concrete | |
| | | Buildings with external load-bearing brick walls; internal - reinforced concrete frame elements | |
| | | Walls made of large blocks (concrete, vibro-brick, or reinforced vibro-brick panels) | |
| | | Reinforced concrete frame with brick filling | |
| 3 | Wooden | One- to two-storey wooden houses (chopped or panel) | EMCA5 |
| | | One- to two-storey wooden frames filled with raw bricks (sinch) | |
| 4 | Concrete | Prefabricated reinforced concrete frame made of linear elements with a welded joint in the zone of maximum effort, or the same with stiffening diaphragms in one direction (framework III of the IIS-04 series and their modifications) | EMCA2 |

| | | Large-panel walls without anti-seismic measures | |
| --- | --- | --- | --- |
| | | Walls of complex construction (with reinforced concrete inclusions); ceilings - prefabricated reinforced concrete | |
| | | Large panel walls | |
| | | Monolithic reinforced concrete frame | EMCA3 |
| | | Prefabricated reinforced concrete frame-braced frame with monolithic nodes, with stiffening diaphragms in two directions or stiffening cores | |
| | | Frame made of spatial elements (volumetric cross) with monolithic knots | |
| | | Frame made of spatial elements (volumetric cross) with monolithic knots | |
| | | Volumetric blocks per room | |
| 5 | Metal frame | Metal frame or frame with diaphragms (bonds) | EMCA6 |

*6) Carefully rewrite section 3. Please make sure that session 3 only contains results and discussion, while data sources and methods should be included in section 2. For example, the number and type of buildings are necessary to perform the risk assessment so they should be introduced before you estimate the risk. Also, 3.1 speaks about asset values but you provide figures in terms of number of buildings, not economic value, which is mentioned in section 3.2.*

Answer: Thank you for your comment. We have revised section 3 and put the data sources and methods in section 2. Also, we have added the figure in terms of economic value which is mentioned in section 3.2 (Fig. 11).

**3. Seismic risk assessment**

[revised manuscript text omitted]

7) *Please make sure the figure captions contain enough explanation for the reader to understand them, and consider merging figures.*

Answer: We have revised the figure captions to have a better explanation

8) *I think your response to the last comment of reviewer #1 is unclear ("The seismic risk map is calculated only considering the ground conditions"). Please specify in the manuscript what you mean, and discuss how your results compare to past risk assessments done in the region.*

Answer: We have specified in the manuscript the idea of compiling the seismic risk map based on the ground conditions and have added the past risk assessments and similar studies done in the region.

[revised manuscript text omitted]

---

## Author Response (AR3)

Dear reviewer,

We appreciate the time and efforts that you put into the review of our paper. We have revised the manuscript based on your comments. Thank you for making our manuscript clearer and more relevant to the scientific community.

Comment: Macroseismic intensity should be indicated using Roman numerals

*Answer: We have indicated the macroseismic intensity with Roman numerals.*

Comment: General seismic zoning maps should be referred to as GSZ-YEARS. It is also advisable to consistently use the same terminology throughout the manuscript, instead of "general seismic zoning map," "seismic zoning," and so forth. The term OSR-2017 should be replaced in all instances with "General Seismic Zoning (GSZ-2017)."

*Answer: Thank you for pointing out our mistake. OSR-2017 is acronym of the project that came based on our native language, so we got used to it. We revised the acronym to GSZ-2017 and used consistent terminology throughout the manuscript.*

Comment: The "Schematic Map of Seismic Intensity in the Territory of the Republic of Uzbekistan" should be titled to be effectively presented in the text comparison with GSZ-2017.

*Answer: We have revised the title*

Comment: 2.2. Paragraph. Please revise the text with greater attention to the sequential presentation. If this section concerns the evaluation and comparison of Seismic Hazard (SH) maps, please include the information on "Regional Seismic Risk Assessment" in a dedicated section.

*Answer: We have revised it with greater attention to the sequential presentation. Also, we have included the new subsection concerning the information on "Regional seismic risk assessment".*

Comment: 95 «The engineering-geological map of the Republic of Uzbekistan is divided by lithologic composition into 14 districts» Please revise the sentence as it seems to describe not 14 regions but the distribution of 14 types of sedimentary rocks across the Republic of Uzbekistan.

*Answer: We have revised the sentence.*

*In the territory of the Republic of Uzbekistan there are 14 typical types of soils: Rock soils; Limestones; Sands and sandstones and others. More detailed information and map are provided in Figure 1.*

Comment: 115 «The seismic risk probability and economic map of the administrative districts of the Republic of Uzbekistan were developed based on the engineering geological conditions and general seismic zoning maps». It is suggested that references to general seismic zoning maps are made clear.

*Answer: We have revised the reference to general seismic zoning maps.*

*rockfalls and soil erosion.*

*The seismic risk probability and economic map of the administrative districts of the Republic of Uzbekistan were developed based on the engineering geological conditions and maps of General Seismic Zoning (GSZ-2017) (Artikov et al. (2020)). Subsequently, seismic vulnerability levels were assessed using the GESI_Program software developed by the RADIUS program of the International Federation of Red Cross and Red Crescent Societies during 1999-2001. The assessment considered various construction materials based on cadastral information, considering the types of buildings and their vulnerability functions. The seismic vulnerability levels of buildings were then evaluated in the districts of the Republic. Considering the ground conditions, the economic map of seismic risk probability in the administrative districts of Uzbekistan was developed, showing the probability of not exceeding 50% within 90 years (in trillion soums).*

Comment: 130 National Building Code No.2.01.03-19 "Construction in Seismic Areas" It would be preferable to present soil classification in a table format for better clarity

*Answer: The Table is comparatively large and takes up a lot of space and we decided not to include it in this paper. Also, since we included this table in some of our previous papers, we thought that including it in every published paper would not be appropriate. However, we cited our previous paper, where we include the table for better clarity:* Ismailov, V. A., Yodgorov, S. I., Khusomiddinov, A. S., Yadigarov, E. M., Botirovich, A. S., & Aktamov, B. U. (2023). New classification of soils by seismic properties for the building code in Uzbekistan. Geomechanics and Geoengineering, 1–21. https://doi.org/10.1080/17486025.2023.2296975

Comment: 140 "Using Fig. 1 of the140 engineering-geological conditions of the territory of the Republic of Uzbekistan…" - requires rewriting for clarity, such as 'Lithological data (Figure 1) were utilised….".

*Answer: We have revised the sentence.*

*Using the lithological data of rocks located within the territory of the Republic of Uzbekistan depicted in Fig. 1, along with the General Seismic Zoning (GSZ-2017) (Fig. 2), we have created a schematic map illustrating seismic intensities across the Republic of Uzbekistan (Fig. 3).*

Comment: 145 "the seismic risk of the city of Tashkent was assessed using a scenario earthquake".

*Answer: Yes, seismic risk of the city of Tashkent was assessed using a scenario earthquake.*

Comment: The assessment of seismic risk for the city of Tashkent, carried out using a scenario earthquake, lacks clarity in the text regarding whether it is part of a seismic risk assessment for the entire territory of Uzbekistan. If it is, a distinct paragraph with a scenario earthquake description and maps is necessary. Otherwise, details of the seismic risk calculation for Tashkent should be included in the Introduction

*Answer: The seismic risk assessment of the city of Tashkent is not part of the seismic risk assessment of the entire territory of the Republic of Uzbekistan. We included data on seismic risk assessment was included in the introduction.*
*In the study by Rashidov et al. (2003), the seismic risk of Tashkent was evaluated using a scenario earthquake. Similarly, in the RADIUS (1992) project, the seismic risk of the city was assessed employing a scenario earthquake. The estimated total damage resulting from this scenario earthquake, encompassing the disruption of life support systems and infrastructure in Tashkent, amounts to approximately 1 billion Uzbekistani soms. (These loss figures are determined based on 1991 prices and are considerably underestimated).*
*Given Tashkent's status as the capital, responsible for a quarter of the country's gross domestic product, the repercussions of an earthquake are poised to impact the entire nation. The potential disruption of numerous international commercial, banking, and insurance networks was anticipated. Human casualties are projected to be significant, with economic recovery likely to span several years. Furthermore, the cessation of industrial production is anticipated to result in losses totaling around 1 billion U.S. dollars. Preliminary calculations indicate that the scenario earthquake could incur damages exceeding 10 billion U.S. dollars, considering the book value of fixed assets determined at 1991 prices. Expert assessments suggest that roughly 80% of communication facilities may remain inoperative for an extended duration, while ongoing construction projects may suffer irreparable damage estimated at approximately 1 billion U.S. dollars.*

Comment: 170 The footnote about 'local building code6' at the bottom of the page is superfluous, as the National Building Code No.2.01.03-19 is already mentioned in the text and referenced.

*Answer: It is superfluous indeed. Thank you for pointing out on this mistake. We removed the footnote.*

---

## Author Response (AR4)

Dear reviewer,

We appreciate the time and efforts that you put into the review of our paper. We have revised the manuscript based on your comments. Thank you for making our manuscript clearer and more relevant to the scientific community.

Comment: Make sure that Table 1 also includes macroseismic intensities expressed in Roman numerals as in the text.

Answer: We have corrected the comments.

**Table 1. Comparison of the ratio of areas with different intensities (based on the MSK-64 macroseismic scale) between two seismic hazard maps, one considering ground conditions and the other not**

|  | V | VI | VII | VIII | IX |
|---|---|---|---|---|---|
| Seismic hazard map | 31,1% | 26,8% | 31,8% | 9,3% | |
| Seismic hazard map with consideration of ground conditions | 16,2% | 39,5% | 27,1% | 10,7% | 6,5% |

Comment: In Figure 7, it is not clear whether the seismic intensity is macroseismic, in which case you should also use Roman numerals.

Answer: We have corrected the comments.

[Figure]

**Fig. 7: Graph of changes in the average degree of damage to individual mudbrick houses depending on the seismic intensity according to different authors.**

Comment: Author's permission is required for changes in Figures 2 and 3.

Answer: We received permission from the authors for the changes in Figure 2. The author of the 3rd picture is us